# Technical note:Evaluation of profile retrievals of aerosols and trace gases for MAX-DOAS measurements under different aerosol scenarios based on radiative transfer simulations

Xin Tian [1,2], Yang Wang [*3#], Steffen Beirle [3], Pinhua Xie [*2,4,5,6], Thomas Wagner [3], Jin Xu [2], Ang Li [2], Steffen Dörner [3], Bo Ren [2,6], Xiaomei Li [2]

1. Information Materials and Intelligent Sensing Laboratory of Anhui Province, Institutes of Physical Science and Information Technology, Anhui University, Hefei, 230601, China;

2. Key laboratory of Environmental Optical and Technology, Anhui Institute of optics and Fine Mechanics, Chinese Academy of Science, Hefei, 230031, China;

3. Max Planck Institute for Chemistry, Mainz, 55128, Germany;

# now at EUMETSAT, Darmstadt. Germany;

4. CAS Center for Excellence in Urban Atmospheric Environment, Institute of Urban Environment, Chinese Academy of Sciences, Xiamen, 361021, China;

5. University of Chinese Academy of Sciences, Beijing, 100049, China;

6. School of Environmental Science and Optoelectronic Technology, University of Science and Technology of China, Hefei, 230026, China;

*Author:* Xin Tian (xtian@ahu.edu.cn)

*Correspondence to:* Pinhua Xie (phxie@aiofm.ac.cn); Yang Wang (y.wang@mpic.de)

**Abstract**: Ground-based Multi-AXis Differential Optical Absorption Spectroscopy (MAX-DOAS) is a state of the art remote sensing technique for deriving vertical profiles of trace gases and aerosols. However, MAX-DOAS profile inversions under aerosol pollution scenarios are challenging because of the complex radiative transfer and limited information content of the measurements. In this study, the performances of two inversion algorithms were evaluated for various aerosol pollution scenarios based on synthetic slant column densities (SCDs) derived from radiative transfer simulations. Compared to previous studies, in our study much larger ranges of AOD and $NO_2$ VCDs are covered. One inversion algorithm is based on optimal estimation, the other uses a parameterized approach. In this analysis, 3 types of profile shapes for

aerosols and $NO_2$ were considered: exponential, Boltzmann, and Gaussian. First, the systematic deviations of the retrieved aerosol profiles from the input profiles were investigated. For most cases, the AODs of the retrieved profiles were found to be systematically lower than the input values, and the deviations increased with increasing AOD. Especially for the optimal estimation algorithm and for high AOD, these findings are consistent to the results in previous studies. The assumed single scattering albedo and asymmetry factor have a systematic influence on the aerosol retrieval. However, for most cases the influence of the assumed SSA and AP on the retrieval results are rather small (compared to other uncertainties). For the optimal estimation algorithm the agreement with the input values can be improved by optimizing the covariance matrix of the *a priori* uncertainties. Second, the aerosol effects on the $NO_2$ profile retrieval were tested. Here, especially for the optimal estimation algorithm, a systematic dependence on the $NO_2$ VCD was found with a strong relative overestimation of the retrieved results for low $NO_2$ VCDs and an underestimation for high $NO_2$ VCDs. In contrast, the dependence on the aerosol profiles was found to be rather low. Interestingly, the results for both investigated wavelengths (360 nm and 477 nm) were found to be rather similar indicating that the differences in the radiative transfer between both wavelengths have no strong effect. In general, both inversion schemes can well retrieve the near-surface values of aerosol extinction and trace gases concentrations.

**1 Introduction**

In recent years, several large-scale aerosol pollution incidents in China (Hu et al., 2014;

Huang et al., 2014; Wang et al., 2014; Zhang et al., 2015) have drawn increasing attention due to their effects on atmospheric visibility and health. Atmospheric aerosols also exert direct and indirect effects on global climate change and radiative balance (Seinfeld and Pandis, 1998; IPCC, 2007). The physical and chemical properties, and the spatial-temporal distributions of aerosols can both affect remote sensing measurements of trace gases in the atmosphere (Seinfeld and Pandis, 1998; Quinn and Coffmann, 1998; Bond et al., 2001; Sheridan et al., 2001). Measuring the optical properties of aerosols, understanding the role of aerosols in atmospheric processes, and assessing the effects of aerosols on remote sensing observations of trace gases are important goals in the study of atmospheric pollution.

The ground-based Multi-AXis Differential Optical Absorption Spectroscopy (MAX-DOAS) technique can be performed with a relatively simple set-up and very low power consumption in the ultraviolet (UV) and visible (Vis) spectral range to synchronously measure the vertical distributions of aerosol optical extinction and concentrations of several trace gases (e.g., $NO_2$, $SO_2$, HCHO, HONO, and CHOCHO) in the troposphere (Hönninger and Platt, 2002; Hönninger et al., 2004; Wittrock et al., 2004; Wagner et al., 2004; Frieß et al., 2006). Spectra of scattered sunlight are measured at different elevation angles (EAs) by the MAX-DOAS instrument. The spectra are analyzed by the DOAS technique (Platt and Stutz, 2008), which makes use of the characteristic "fingerprint" absorptions of the different trace gases with respect to a reference spectrum taken for zenith. The results of the spectral fitting process are the so-called differential slant column densities (DSCDs) of the trace gases and the oxygen collision

complex ($O_2$-$O_2$ or $O_4$), with the DSCD defined as the difference between the trace-gas concentration integrated along the effective light path and the corresponding integrated trace-gas concentration in the zenith sky reference spectrum. The MAX-DOAS technique basically utilizes the EA dependence of differential absorption structures of $O_4$ to derive the vertical distribution of the aerosol extinction (Wagner et al., 2004; Frieß et al., 2006). The vertical profiles and vertical column densities (VCDs) of trace gases can be retrieved from the EA dependence of DSCDs using also the result of the aerosol profile inversion from MAX-DOAS (Irie et al., 2008, 2009; Li et al., 2010; Clémer et al., 2010; Hartl and Wenig, 2013; Hendrick et al., 2014; Vlemmix et al., 2015; Frieß et al., 2006).

Recent research on MAX-DOAS has focused on the following aspects: (1) profile inversion algorithms (Hönninger and Platt, 2002; Wagner et al., 2004; Frieß et al., 2006, 2011; Clémer et al., 2010; Hay, 2010; Vlemmix et al., 2011; Yilmaz, 2012; Hartl and Wenig, 2013; Holla, 2013; Wang et al., 2013a, b; Zielcke, 2015; Bösch et al., 2018; Beirle et al., 2019; Friedrich et al., 2019; Spinei et al., 2019; Frieß et al., 2019); (2) long-term observation of trace gases and aerosols (e.g., Irie et al., 2008a; Roscoe et al., 2010; Li et al., 2013; Ma et al., 2013; Pinardi et al., 2013; Hendrick et al., 2014; Kanaya et al., 2014; Wang et al., 2014; Chan et al., 2015; Tian et al., 2017, 2018; Wang et al., 2017a); (3) cloud identification and data correction (Gielen et al., 2014; Wagner et al., 2014, 2016; Wang et al., 2014); and (4) satellite and model data validation (e.g., Halla et al., 2011; Ma et al., 2013; Pinardi et al., 2013; Chan et al., 2015; De Smedt et al., 2015; Vlemmix et al., 2015; Jin et al., 2016; Drosoglou et al., 2017; Wang et al., 2017b;

Boersma et al., 2018; Liu et al., 2018). In this study we focus on the first aspect. At present, algorithms for the retrieval of vertical profiles from MAX-DOAS measurements can be separated into optimal estimation methods (OEMs) (Rodgers, 2000) and parameterized algorithms, which describe the shapes of atmospheric profiles

with a limited set (usually 2 to 3) of parameters. In Frieß et al. (2019), different MAX-DOAS inversion schemes have been compared for synthetic input data for AODs up to 1 (plus a fog and two cloud scenarios). Given the importance and complexity of the aerosol effects on the atmospheric radiative transfer, it is also important to study the impact of heavy aerosol loads on the MAX-DOAS inversion algorithm.

Here, we compare the aerosol and trace gas profiles retrieved from MAX-DOAS by two inversion algorithms (PriAM and MAPA, for details see below) with the input values (used as input for the DSCD simulations) for different aerosol scenarios. We also investigate the effects of the aerosol extinction and optical properties, including single-scattering albedo (SSA) and the asymmetry parameter (AP), on the aerosols

profiles retrieved by PriAM in the UV and Vis.

This manuscript is organized as follows. Section 2 briefly describes the basic settings for the aerosol and $NO_2$ profile inversions and for the tests of the profile comparisons. The analysis strategy of this study is presented in Section 2.1. The model scenarios and radiative transfer model (RTM) settings are specified in Section 2.2. The 2 profile

retrieval algorithms (PriAM and MAPA v. 0.98) are described in Section 2.3. The effects of aerosols on the profile retrievals are discussed in Section 3.

## 2 Basic settings and tests

### 2.1 Analysis strategy

The analysis strategy of this study is depicted in **Fig. 1**. A set of atmospheric scenarios variations of (orange box on the left side), including the viewing geometries, single-scattering albedos, and asymmetry parameters, was used to simulate the SCDs of traces gases and $O_4$, which will be described in detail in Section 2.2. The first step was to quantitatively evaluate the effect of different aerosol loads on the aerosol inversion (The upper part of the Fig.1). For that purpose the simulated $O_4$ DSCDs were used as input for the aerosol profile retrievals. The retrieved and input aerosol profiles were then compared in order to characterize the effect of the aerosol properties (in particular the AODs) on the retrieved aerosol profiles. The second step was to quantitatively evaluate the effect of different aerosol loads on the trace gas inversion (the bottom half of the Fig.1). For the trace gas retrievals, we apply 2 retrieval strategies where either the retrieved (S1, red box in the lower half of Fig.1) or the input (S2, red box in the lower half of Fig.1) aerosol profile is used.

### 2.2 RTM parameters

Before the effects of different aerosol loads on the retrieval of aerosol and trace gas profiles were analyzed, some basic parameters were prescribed for simulating the $O_4$ and trace gas SCDs for the 'assumed input profiles' in the RTM. In this study, the SCIAMACHY radiative transfer model (SCIATRAN) (version 2.2, Rozanov et al., 2005) is used in the forward model calculations. Here it is important to note that while

SCIATRAN is also used in PriAM, in the MAPA algorithm a different RTM (MCARTIM, Deutschmann et al., 2011) is used. The differences of the simulated $O_4$ DSCDs by both models are discussed in section 3.1.2.

SCIATRAN models radiative transfer processes in the terrestrial atmosphere and ocean in the spectral range from the ultraviolet to the thermal infrared including all significant radiative transfer processes, e.g., the Rayleigh scattering, scattering by aerosol and cloud particles, and absorption by gaseous components and aerosols (Rozanov et al.,2014). The RTM used in this section was SCIATRAN version 2.2. The Monte Carlo Atmospheric Radiative Transfer Inversion Model (McARTIM) is a full spherical Monte Carlo model without polarization (Deutschmann et al., 2011). In a recent intercomparison activity within the project FRM4DOAS (https://frm4doas.aeronomie.be/), in general very good agreement (deviations up to a few percent) between MCARTIM and SCIATRAN version 2.2 was found with the largest deviations for cases with fog or shallow box profiles (Frieß et al., 2018). It should also be noted that the agreement between MCARTIM and SCIATRAN v3.0 is better than with SCIATRAN v2.2. The differences between $O_4$ DSCDs simulated by SCIATRAN and MCARTIM are further investigated in Section 3.1.2.

Retrievals based on synthetic SCDs for various viewing geometries in the UV and Vis were performed. The dependencies on the retrieval parameters and settings, different measurement viewing geometries, and different aerosol and trace gas profile shapes were identified by comparison of the results to those of the standard settings. As standard settings we chose wavelengths of 360 nm and 477 nm, elevation angles of 1°,

2°, 3°, 4°, 5°, 6°, 8°, 15°, 30°, and 90° (the same as the settings in the CINDI 2 campaign,

Kreher et al., 2020). In the real atmosphere, a large variability of aerosol and trace gas

profiles exists. However, we had to limit our profile shapes to typical profile shapes,

which occur in the atmosphere. In this study, three different profile shapes were used,

which are Exponential, Boltzmann, and Gaussian profile shapes:

a) Exponential profiles: such profiles are typical if the emissions mainly occur at the

surface. During transport to higher layers the concentration systematically decreases

with altitude. The scale height depends on the atmospheric lifetime and the vertical

transport time. The description for Exponential functions of altitude z as follows:

Exponential: $f_E(z) = A_E(h_E) \times \exp(\dfrac{-z}{h_E})$ with scale height $h_E$,

b) Boltzmann profiles: Such profiles represent situations, for which a layer is quickly

mixed (compared to the lifetime of the species), and there is a barrier for further

upwards transport above that layer. Such situations typically occur for well mixed

boundary layers. The description for Boltzmann functions of altitude z as follows:

Boltzmann: $f_B(z) = \dfrac{A_B(h_B)}{1 + \exp(\dfrac{-(z - h_B)}{0.3})}$ with effective profile height $h_B$.

c) Gaussian profiles: in our study these profiles describe elevated layers. Such profiles

represent situations with long range transport of pollutants, which typically occurs

above the boundary layer. Elevated profiles might also occur for aerosols and trace

gases which are secondary formed, while air is transported upwards. The description

for Gaussian functions of altitude z as follows:

Gaussian: $f_G(z) = A_G(h_G, \sigma) \times \exp(\dfrac{-(z - h_G)^2}{2\sigma^2})$    with peak height $h_G$, and the full

width at half maximum (FWHM) $\sigma$ .

The normalization factors $A_E$, $A_B$, and $A_G$ are determined by numerical integration from 0 to 4 km altitude such that the integrals of $f_E$, $f_B$ and $f_G$ equal 1, respectively. The value above 4 km altitude is set to 0.

For RTM calculations, vertical profiles of the aerosol extinction $\varepsilon$ and NO$_2$ concentration c are generated by multiplying f with the respective *a priori* column:

$$\varepsilon = f(z) \times \tau ,$$

$$c = f(z) \times VCD .$$

**Figure S1** displays the corresponding vertical profiles for the different shapes. **Table 1**
lists the parameters used for RTM, including solar/viewing geometry, *a priori* AOD/VCD, and parameters for the different profile shapes. The profile shape scenarios are introduced in detail in Section 3.1.

## 2.3 Description of the retrieval algorithms

The retrieval algorithms used in the comparison were PriAM and MAPA, as listed in **Table 2**.

### 2.3.1 PriAM algorithm

The PriAM profile inversion algorithm of aerosol extinction and trace gas concentration
developed by the Anhui Institute of Optics and Fine Mechanics, Chinese Academy of Sciences (AIOFM, CAS), in cooperation with the Max Planck Institute for Chemistry (MPIC) (Wang et al., 2013a and b, 2016), is based on the nonlinear optimal estimation

method using the Levenberg–Marquardt modified Gauss–Newton numerical iteration procedure (Rodgers, 2000). PriAM uses the radiative transfer model (RTM) SCIATRAN version 2.2 (Rozanov et al., 2005) to calculate the weighting functions and other simulated quantities. PriAM consists of a 2-step inversion procedure. In the first step, aerosol extinction profiles are retrieved from the dependence of the $O_4$ DSCDs on elevation angle. The single-scattering albedo and asymmetry parameters have to be prescribed for the aerosol retrieval, e.g. based on other auxiliary measurements. Subsequently, profiles of the trace gas number density are retrieved from the respective DSCDs in each MAX-DOAS elevation angle sequence (Wang et al., 2017). In order to avoid negative concentrations in the retrieved results (which are not possible in the actual atmosphere), the retrievals are performed in logarithmic space. Here it should be noted that since the distribution probabilities of the retrieved profiles around the *a priori* profiles become asymmetric due to the inversion in logarithmic space, the sensitivity of the inversion to large values is greater than the sensitivity in linear space (Wang et al., 2019). PriAM can retrieve trace gas and aerosol profiles on any arbitrary vertical grid. In this study, vertical layers with 200-m resolution in the altitude range below 4.0 km were used.

### 2.3.2 MAPA algorithm

The Mainz profile algorithm (MAPA) is a parameter-based inversion method using a Monte Carlo (MC) approach developed by the Max Planck Institute for Chemistry (MPIC) (Beirle et al., 2019). Here we use MAPA v0.991, which is basically the same

algorithm as described in Beirle et al., 2019 (v0.98), with only slight differences in the flagging procedure.

The radiative transfer model used in MAPA (McArtim, Deutschmann et al., 2011) for calculating each parameter in the lookup tables (LUTs) is a full spherical Monte Carlo model. MAPA also comprises a 2-step inversion procedure. First, the aerosol profile is retrieved based on $O_4$ DSCDs. In this step, other input parameters include the errors of the $O_4$ DSCD, the $O_4$ VCD and information about the viewing geometry (elevation angle (EA), solar zenith angle (SZA), and relative azimuth angle (RAA)). Next, the trace gas profiles are retrieved based on the aerosol profiles derived in step 1 and the trace gas DSCDs (and their errors). Three parameters (layer height, profile shape, and integrated column (AOD or VCD)) of the aerosol and trace gas profiles are derived in the inversion. The final profiles are weighted averages of the best matching profiles for the given trace gas DSCDs. The details of MAPA can be found in Beirle et al. (2019). It is worth noting that the maximum AOD in MAPA is 3, since higher AODs were not included in the RTM look-up table; therefore, only aerosol scenarios with AOD $\leqslant$ 3 were included in this study for MAPA.

## 3 Results and discussion

In order to simulate the effects of different aerosol loads on the MAX-DOAS profile inversion algorithms, the aerosol and trace gas profiles were set up with 5 AOD and 5 VCD values as presented in **Tables 1** and different height parameters as shown in **Table 1**). The fitting error for all $O_4$ DSCDs is set as $0.03 \times 10^{43}$ molecules$^2$ cm$^{-5}$, and that for

$NO_2$ DSCDs to 1% of the $NO_2$ DSCDs in the PriAM and MAPA retrievals.

In order to limit the number of investigated profiles, first a sensitivity study with PriAM was carried for the selected profile shapes in Table 1 (these best represent the variety of realistic profile shapes). Based on the result shown in **Figs. S2 to S4** it turned out that one height parameter is mostly representative for the parameterization with Gaussian and Boltzmann profiles. For the exponential profiles, two height parameters were chosen, because for both height parameters systematically different results were obtained: when the scale heights of the exponential profiles are low, the retrieved profiles are close to the input profiles. But for high scale height, the retrieval underestimates the scale height of the exponential profiles.

The settings of the 4 chosen profile shapes are listed in **Table 1**. The 4 profiles are exponential profiles with scale heights of 0.5 km and 1.0 km, respectively, Gaussian profiles with the peak height at 1.0 km and FWHM of 0.5 km, and Boltzmann profiles with a height of 1.5km.

A similar sensitivity study was also performed for the trace gas profiles. The results of the sensitivity analysis (**Figs. S5 to S7**) for $NO_2$ profiles are consistent with the findings for the aerosol profiles. Thus the settings of the $NO_2$ profile shapes for all further tasks are the same as for the aerosol profile in **Table 1**.

**3.1 Aerosol results**

In this Section the effect of different AOD on the retrieval of aerosol profiles are presented for a scenario with SZA = 60°, RAA = 120°, SSA = 0.9, and AP = 0.72. Note

that similar results were found for different scenarios for both PriAM and MAPA. The effects of the different SZA (20°, 40°, 60°, 80°) and RAA (30°, 60°, 120°, 180°) are basically the same. But here it is important to note that in the real atmosphere, very different phase functions might occur, and especially for small RAA stronger systematic deviations might occur. Here only the result for SZA = 60° and RAA = 120° was shown. In addition, the effects of SSA and AP are further explored in Section 3.1.4.

### 3.1.1 Aerosol profile comparison of PriAM and MAPA

**Fig. 2** shows the comparison of the input aerosol profiles and the corresponding profiles retrieved by PriAM and MAPA for the different profile shapes and AOD scenarios. Note the relative and absolute deviations between the retrieved and input aerosol profiles are shown in **Fig. S8** and **Fig. S9**, respectively. The results reveal that both PriAM and MAPA can overall well retrieve the 4 different profile shapes (**Fig.2**). Similar results were derived for the aerosol retrievals at 360 nm and 477 nm. However, also absolute deviations are found, which increase as the AOD increases. The magnitudes of the absolute deviations between the retrieved and input aerosol profiles for PriAM were smaller than for MAPA at AOD <1.0. For the exponential profiles with scale heights of 1.0 km, the magnitudes of the absolute deviations between the retrieved and input aerosol profiles were the smallest among the 4 profile shapes. The derived profiles for exponential profiles with scale heights of 0.5 km were in better agreement for PriAM than for MAPA. The maximum magnitudes of the absolute deviations primarily occurred at heights < 1.0 km. Here it is interesting to note that the parameterization used in MAPA does not include pure exponential profiles, but only

combined profiles with a (shallow) box profile at the bottom and an exponential profile on top. This limitation can explain the large magnitudes of the absolute deviations for MAPA retrievals especially at low altitudes. For the Boltzmann-shaped profiles, the height around which the maximum magnitudes of the absolute deviations for PriAM and MAPA often occurred was 1.0 km, but for AOD > 1.0, the magnitudes of the absolute deviations for PriAM in the 200-m layer were greater than for MAPA. In brief, the concentrations of the 200–400 m layer retrieved by PriAM were moderately larger than those of the input profiles for AOD > 1.0. Thus, better agreement for the Boltzmann profiles was found for MAPA than for PriAM. For the Gaussian-shaped profiles, both PriAM and MAPA could well retrieve the lifted layer. The width of the lifted layer retrieved by MAPA was close to the truth, although the aerosol extinction was underestimated. PriAM underestimated the width of the lifted layer, but the aerosol extinction was closer to the input value (**Fig. 2**). The height at which the maximum magnitudes of the absolute deviations for the Gaussian-shaped profiles mainly occurred was 1.5 km. The relative deviations between the retrieved and input aerosol profiles for different AOD scenarios are similar for the same retrieval algorithm with the magnitude of the relative deviations  for AOD >1.0 obviously greater than for AOD < 1.0. But the magnitude of the relative deviation does not increase with the increase in AOD.

**3.1.2 Differences of the $O_4$ SCDs simulated by SCIATRAN and MCARTIM**

PriAM and MAPA use different RT models, which might partly explain systematic differences. In order to quantify the impact of the differences between SCIATRAN and

MCARTIM, $O_4$ DSCDS calculated by MCARTIM are compared to those calculated by SCIATRAN for selected cases.

Because the aerosol properties used in the MAPA LUT (SSA = 0.95 and AP = 0.68) are different from those used for the simulations of the $O_4$ DSCDs by SCIATRAN (SSA = 0.90 and AP = 0.72), two sets of $O_4$ DSCDs for SSA and AP (SSA = 0.90 or 0.95 and AP = 0.72 or 0.68) were simulated by MCARTIM.

The comparison results for the $O_4$ DSCDs (**Fig. S10**) show that differences between the SCIATRAN and MCARTIM simulations using the same SSA and AP of 0.9 and 0.72, respectively, are up to 9%. If also different aerosol properties were used, these differences increased further.

In the next step, the differences of the retrieval results for the different input DSCDs are investigated. The corresponding results are shown in **Fig. S11.** Interestingly, it is found that the exact choice of the aerosol optical properties has only a small influence on the results.

Using McArtim for the calculation of synthetic DSCDs, i.e. consistent RTM in forward model and inversion, results in much better agreement, in particular for low AOD. Thus, the large relative deviations for MAPA seen in **Fig. 7** are partly explained by the differences in RTM. For the Gaussian profiles, the larger differences at high AODs occur due to the obvious overestimation of the width of the lifted layer.

.

**3.1.3 Sensitivity study of the *a priori* profile and the *a priori* profile covariance matrix**

In order to improve the profile inversion accuracy for high AODs, the influence of the *a priori* profile and the *a priori* profile covariance matrix (Sa) was examined for PriAM. Here it should be noted that an exponential shape with an AOD of 0.2 and a scale height of 1.0km was used as universal *a priori* profile in this study. In order to investigate the importance of the *a priori* profile for the aerosol profile retrieval, the influence of the *a priori* profile was analyzed by changing the *a priori* profile to different aerosol profile shapes. Also, in addition to an AOD of 0.2 a second AOD value of 2.0 is used. The *a priori* profiles used in the sensitivity test are presented in **Fig. 3**. Here it should be noted that either the exponential profile shapes (universal *a priori* profile in PriAM in this study) or the same profile shapes (Boltzmann or Gaussian) as the input profiles are also used as *a priori* profiles (referred to as 'corresponding *a priori* profiles' in the following). The comparison of the retrieved profiles using the different *a priori* profiles with the input profiles are shown in **Fig. 4**. It is found that the inversion results of the aerosol profile were slightly improved by changing the *a priori* profiles to the corresponding profile shapes, and that for the high AOD scenarios the inversion results were further improved by increasing the AOD of the corresponding *a priori* profile (**Fig.4**). However, increasing the AOD of the universal (exponential) *a priori* profile exhibited only little effect on the inversion results of the Boltzmann and Gaussian shapes. It is worth noting that when the input aerosol extinction coefficient was small, the use of *a priori* profiles with high AOD often yielded unrealistic results.

We also investigated the retrieval results in a perfect scenario in which the *a priori* profile agrees with the input profile. The results are presented in **Fig. 5**. The results

show that the retrieved aerosol profiles are basically the same as the input profiles, and the relative deviation is less than 0.05% (**Fig. S14** of the supplement). This sensitivity study shows that a) PriAM is implemented in a proper way and b) improved retrieval results can be obtained with improved *a priori* profiles. This provides a possibility for real measurements to obtain more accurate aerosol profiles if independent information on the *a priori* profiles is available, e.g. from Lidar observations and sun photometers.

The Sa is the covariance matrix of the *a priori* profile (N×N), and its diagonal elements are the square of the *a priori* state uncertainties with the off-diagonal elements calculated from the Gaussian function with the correlation length of 0.5 km (Frieß et al., 2006). The universal *a priori* settings of Sa in this study was such that the diagonal elements decreased exponentially with height. As a consequence, the smaller the Sa values, the more the inversion results depends on the prior state vector. The diagonal elements of Sa for the aerosol profile were set as the square of the *a priori* profile uncertainty. The standard settings for the *a priori* profile uncertainty were 10% of the *a priori* profile. To describe this ratio, a new symbol (Sa_ratio) is introduced (see Table 4). The effect of different Sa values on the retrieval of the 4 aerosol profiles was studied, and the results for an AOD of 5.0 are shown in **Fig. 6** (The profile results show that the deviation magnitudes of absolute deviations (**Fig.S9)** between the retrieved and input profile increase with the increase of the AOD, so a high AOD of 5.0 was selected to show the impact for an extreme case). The 4 Sa_ratio were set to 6%, 10%, 20%, and 50%. For the exponential profiles with a scale height of 0.5 km, the correlation coefficient between the retrieved and input aerosol profiles decreased with increasing Sa. But the correlation coefficient could be improved by increasing the Sa values for other profiles (exponential profiles with a scale height of 1.0 km, Gaussian and

Boltzmann aerosol profile shapes). In particular, the retrieved surface extinctions and scale heights could be improved by increasing the Sa. This is due to the fact that the biases towards the *a priori* profiles are reduced with increasing Sa values. When the Sa values were too large, however, the retrieved aerosol profiles in the upper layer (approximately above 2.0 km) were more unstable. The highest correlation coefficient was found when the diagonal elements of Sa were set to the square of 20% of the *a priori* profile for the Boltzmann profiles and exponential profiles with a scale height of 1.0 km at AOD of 5.0, with the smallest root-mean-square deviation (RMSD) of 0.54 and 0.50 (averaged of 360nm and 477nm for each shape), respectively. For the Gaussian profile, the correlation coefficient was highest with the diagonal elements of Sa in 50% of the *a priori* profile. The smallest averaged RMSD of 0.55 was also found for this scenario with values of 0.58 at 360nm and 0.52 at 477nm, respectively.

### 3.1.4 Comparison of retrieved and input $O_4$ DSCD for PriAM and MAPA

The modeled $O_4$ DSCDs corresponding to the aerosol profiles retrieved by PriAM and MAPA were compared to the input $O_4$ DSCDs simulated by the RTM. The comparison results are shown in **Fig. 7** for the different aerosol profile shapes and the 5 AOD values for 360 and 477 nm. Note that only the results for AOD $\leqslant$ 3.0 were derived from MAPA. Also the slopes, intercepts, and correlation coefficients are shown in **Fig. 7**. The correlation coefficients ($r^2$ values) were > 0.99 for both the PriAM and MAPA results. Also the slopes are very close to unity. Therefore, it can be concluded that the discrepancies of the retrieved aerosol profiles from the input profiles were not caused by failed convergences of the retrievals but must be related to systematic performances of the inversion algorithms in solving the ill-conditioned problem or RTM differences.

### 3.1.5 AOD comparison of PriAM and MAPA

**Fig. 8** shows the deviations of the AODs retrieved by PriAM and MAPA with the input AODs for the 4 selected aerosol profiles and 5 AOD values at the wavelength of 360nm and 477nm. Both PriAM and MAPA underestimate in general the input AODs at these two wavelengths. For the exponential aerosol profiles with a scale height of 0.5 km, the magnitude of the relative deviations of the retrieved AODs by PriAM and MAPA compared to the input AODs are less than 20% for most AODs. In contrast, much worse agreement is found for the exponential profiles with scale height of 1.0 km. The magnitude of the relative deviations between the retrieved and input profiles are > 20% and are similar for PriAM and MAPA. The main reason is that the retrieved scale height for exponential profile of 1.0km by PriAM and MAPA is significantly lower than the input profile. Especially for low AOD, the AODs retrieved by PriAM are closer to the input AODs than those retrieved by MAPA. Part of the systematic underestimation of the MAPA AODs for exponential profiles is probably caused by the differences of the RTM (SCIATRAN v2.2) and settings (SSA=0.9, AP=0.72) used for the simulation of the input $O_4$ DSCDs and for the MAPA algorithm (MCARTIM, SSA=0.95, AP=0.68), see **Fig. S11**. Another reason might be, as mentioned in section 3.1.2, the limitation to accurately describe purely exponential profile shapes. The different incorporated methods for providing the *a priori* information is also a potential reason for the differences between the two retrieval algorithms. Prescribed *a priori* profile and the *a priori* covariances are used in PriAM, while *a priori* assumptions are incorporated in

MAPA in the form of prescribed profile shapes by the chosen parameterization.

For the Boltzmann and Gaussian profile shapes the relative deviations between the retrieved and the input AODs increased with increasing AODs for both PriAM and MAPA. The largest relative deviations magnitude are >50% for large AODs.

**3.1.6 Effect of single scattering albedo and asymmetry parameter used in the inversion of the retrieved aerosol profiles**

The effects of single-scattering albedo (SSA) and asymmetry parameter (AP) used in the forward model of the aerosol profile inversion by PriAM were examined. First, a single aerosol profile was used to simulate the $O_4$ DSCDs for different SSA (0.8, 0.9, 1.0) and AP (0.68, 0.72) values (See **Table 1**). Next, the simulated $O_4$ DSCDs were used to retrieve the aerosol extinction profiles by PriAM using the "correct" SSA and AP values (hence, the same values as they were applied in the corresponding $O_4$ DSCD simulations). The retrieved aerosol profiles for all SSA and AP values are shown in **Fig. 9**. These results reveal that especially for low AODs the retrieved aerosol extinction profiles are very consistent for these scenarios. The relative and absolute deviations of the resulting aerosol extinction profiles to the input profiles are presented in **Fig. S16 and Fig. S17**. The results are consistent with those presented in **Figs.2 and S9**. It is worth noting that the magnitude of the relative deviation for the Boltzmann aerosol profiles retrieved for SSA = 0.9 and AP = 0.72 was smaller than for the other scenarios.

In the next step, the effect of incorrect SSA and AP values **(Table 3)** on the aerosol profile inversion was studied using the PriAM standard settings with SSA = 0.9 and AP = 0.72 for the simulation of the $O_4$ DSCDs. The comparison of the retrieved profiles

from the profiles with the incorrect SSA and AP values are presented in **Fig. 10**. It was found that when the SSA was smaller than the input value, the retrieved extinction profiles were larger than the input profiles and vice versa. It is worth noting that the result at 0 km is found to be opposite. For the AP the opposite dependency was found.

The effect of incorrect SSA and AP values on the aerosol profiles retrieved by PriAM increased with increasing AOD with the absolute deviations of the extinction coefficient increasing from 0.01 to 1.5 $km^{-1}$ as the AOD increased from 0.1 to 5.0.

### 3.2 NO$_2$ results

First, the effects of different aerosol extinction profiles on the trace gas profile inversion for 5 NO$_2$ VCDs ($0.1 \times 10^{16}$, $0.3 \times 10^{16}$, $1.0 \times 10^{16}$, $3.0 \times 10^{16}$, and $10.0 \times 10^{16}$ molecules $cm^{-2}$) were examined using aerosol profiles with 4 AODs (0.3, 1.0, 3.0, and 5.0) (AOD = 5.0 was not included for MAPA). Two strategies (either the retrieved (S1) or the input (S2) aerosol profiles served as input for the retrievals of the NO$_2$ profiles)

were employed to retrieve the NO$_2$ profiles (see **Section 2.1**). Here, as for the aerosol inversions, also the scenario with SZA = 60°, RAA = 120°, SSA = 0.9, and AP = 0.72 was used. For the NO$_2$ profiles, the exponential profile shape with a VCD of $1.0 \times 10^{16}$ molecules $cm^{-2}$ was utilized as the universal *a priori* profile for PriAM.

**3.2.1 Comparison of NO$_2$ profiles retrieved by PriAM and MAPA**

**Fig. S20 and Fig. S21** shows the relative and absolute deviations of 4 typical NO$_2$ profiles retrieved by PriAM and MAPA using S1 with the input NO$_2$ profiles for the

Boltzmann aerosol profile shapes with 3 AODs (0.3, 1.0, and 3.0) and 5 VCD values. And the comparison result is shown in **Fig. 11.** The results reveal that the magnitude of absolute deviations between the $NO_2$ profiles retrieved by both PriAM and MAPA and the input $NO_2$ profiles are similar and relatively small, despite the differences in level

of agreement of the aerosol inversion. For the same aerosol conditions, the magnitude of the absolute deviations between the retrieved $NO_2$ profiles and the input values increase with increasing $NO_2$ VCDs. However, the magnitude of the relative deviations stays constant (**Fig. S20)**. It is worth noting that the magnitude of the relative deviations between the retrieved $NO_2$ profiles and the input values for low $NO_2$ VCDs was

significantly higher than for high $NO_2$ VCDs for an AOD of 3.0. For the same aerosol conditions, the systematic deviations between the retrieved $NO_2$ profiles and the input values increase with increasing $NO_2$ VCDs, while the magnitude of the relative deviations increases slightly with the increase of AOD for the same $NO_2$ VCD. The largest deviation magnitudes between the retrieved $NO_2$ profiles and the input $NO_2$

profile for the exponential $NO_2$ profiles with scale height of 0.5 km were mainly found below 1.0 km. The largest deviation magnitudes between the retrieved $NO_2$ profile and the input $NO_2$ profile appeared below 2.0 km for the other three profile shapes, with the maximum deviation magnitude occurring at 1.0 km and 0.2 km. The reason for this finding is that the sensitivity above 1.0 km gradually decreases with increasing AOD,

making it impossible to correctly retrieve the $NO_2$ values at high altitudes. The smoothing effect of PriAM overestimates the $NO_2$ concentrations around 500 m to compensate for the underestimation of the $NO_2$ concentrations above 1.0 km. In other

words, PriAM yields another solution for the ill-conditioned problem in order to achieve convergence between the retrieved and measured SCDs under the control of the *a priori* profile and its covariance.

In the real atmosphere, the profiles of aerosols and $NO_2$ are often quite different.

Therefore, the effect of 4 typical aerosol profile shapes on the retrieval of Boltzmann $NO_2$ profiles by PriAM and MAPA using S1 with 3 AODs (0.3, 1.0, and 3.0) and 5 VCD values was further studied. The results showed that the relative and absolute deviations (**Figs.S22** and **Fig.S23**) between the Boltzmann $NO_2$ profiles retrieved for the 4 aerosol profile shapes and the input $NO_2$ profiles was basically the same, which

means that the influence of the aerosol profile shapes on the retrieval of the $NO_2$ profiles is small.

The $NO_2$ profiles for the 5 VCDs retrieved for scenarios S1 and S2 by PriAM were further compared with the input $NO_2$ profiles for the 4 AOD conditions (0.3, 1.0, 3.0, and 5.0) (**Fig. 13**). The magnitude of the absolute deviations between the retrieved $NO_2$

profiles using S1 and the input values were smaller than those for scenario S2, mainly because the retrieved scale heights for the S1 inversions were closer to the input scale height (**Fig. S25** of the supplement). An interesting phenomenon was the occurrence of some singular values (outliers which deviate from the true values in some layers) in the upper layers of the retrieved profiles for low $NO_2$ VCDs (mainly for $NO_2$ VCD $< 1 \times$

$10^{16}$ molecules cm$^{-2}$). The $NO_2$ profiles retrieved for scenario S1 were more stable than the profiles for scenario S2, with fewer singular values. When the AOD was large but the $NO_2$ VCD was small, the magnitude of the absolute deviations of the $NO_2$

number density at high altitudes was rather large, mainly because the lack of upper-level information for the $NO_2$ profiles made the inversion results more dependent on the *a priori* profile. When the VCD increased, although the box-AMF at high altitudes was small, the $NO_2$ number density at high altitudes also contribute to the SCDs due to

the high $NO_2$ VCD. Thus, when the AOD was large, the value at high altitudes of the $NO_2$ profile can be better retrieved for increased $NO_2$ VCDs.

The smaller the covariance matrix of the *a priori* profile (Sa), the more the retrieved profile depends on the *a priori* profile, which determines the degree to which the retrieved profile deviates from the *a priori* profile. As standard value of the Sa diagonal

elements for retrieval of $NO_2$ profiles, we used the square of 50% of the *a priori* profile. And an *a priori* profile of exponential shape is used for $NO_2$ retrieval (shown in **Fig. 14**), which may cause the great difference between the retrieved and input $NO_2$ profile, especially for the Gaussian and Boltzmann $NO_2$ profiles. In order to reduce the occurrence of single outliers in the upper layer of the $NO_2$ profile, the Sa was reduced,

thus making the retrieved profile more dependent on the *a priori* profile. The effect of the Sa reduction on the retrieval of the 4 $NO_2$ profile types was examined for AODs of 0.3 and 5.0 (**Fig. 15**). The Sa reduction increased the stability of the $NO_2$ profile retrievals for low $NO_2$ VCDs while simultaneously increasing the retrieved scale height. The increase of Sa for high AOD conditions did not improve the inversion results but

instead increased the occurrence of single outliers. For low $NO_2$ VCDs, the overestimation of the $NO_2$ profile above 2.0 km can be explained by the higher values of the *a priori* profile at the upper layers, because when the AOD is large, the

information content for the $NO_2$ distribution at upper layers is very sparse, and the inversion results mainly depend on the *a priori* profile.

We also investigated the retrieval results if exactly the a priori profiles were used as input profiles. The results are presented in **Fig. 16**. In contrast to the aerosol inversion, here for some scenarios substantial differences are found, which in general increase with increasing $NO_2$ VCD and AOD. The smallest deviations are found for exponential and Boltzmann profiles, whereas for Gaussian profiles larger differences are found. The magnitude of the relative deviation increases from 20% to 50% with the $NO_2$ VCD increasing from $1\times10^{14}$ to $10\times10^{16}$ molecules $cm^{-2}$ (**Fig. S28**). It is important to note that the relative deviations for the retrieved $NO_2$ profile by using both the aerosol and $NO_2$ *a priori* profiles as input profiles are less than those if only the aerosol a priori profile is used as input profile (PriAM by S2). This finding also provides guidance for gas inversions in the real atmosphere, if the aerosol and gas profiles can be provided as the *a priori* profile by other monitoring techniques, the inversion results of MAX-DOAS will be more accurate.

### 3.2.2 Comparison of the retrieved $NO_2$ DSCD by PriAM and MAPA and the input $NO_2$ DSCD for scenario (S1)

The $NO_2$ DSCDs retrieved by PriAM and MAPA for scenario S1 were compared with the input $NO_2$ DSCDs for 4 AOD scenarios and 5 VCDs, as shown in **Fig. 17**. The correlations between the $NO_2$ DSCDs retrieved by PriAM and the input values were similar, and for both algorithms values very close to 1.0 were found. Also for the slopes

values close to 1.0 were found.

### 3.2.3 Comparison of the NO₂ VCDs retrieved by PriAM and MAPA

The NO₂ VCDs retrieved by PriAM and MAPA were compared with the input NO₂

VCDs for 3 AOD scenarios and 5 VCDs, as shown in **Fig. 18**. The NO₂ VCDs were

retrieved for PriAM for scenarios S1 and S2, and for MAPA for scenario S1. The VCDs

retrieved by MAPA were closer to the input VCDs than those retrieved by PriAM. The

retrievals of NO₂ VCDs by MAPA and PriAM were only slightly affected by the AOD.

However, especially for PriAM, a strong and systematic dependence of the relative

deviations on the NO₂ VCD was found for all profile shapes. While for small NO₂

VCDs the retrieved VCDs systematically overestimate the true NO₂ VCDs (by up to

60% for PriAM), for large NO₂ VCDs a systematic underestimation is observed (up to

-20%). For Gaussian and Boltzmann profiles the deviations are larger than for the

exponential profiles. Best agreement is found for NO₂ VCDs around $1 \times 10^{16}$ molecules

cm$^{-2}$. Here it should, however, be noted that while for low NO₂ VCDs the magnitude of

the relative deviations are large, the magnitude of the absolute deviations are rather

small.

### 3.3 Discussion

In this section we discuss the most important findings of our investigations and compare

them to the results from earlier studies. Especially Bösch et al. (2018) and Frieß et al.

(2019) investigated the sensitivity of the MAX-DOAS inversion results using synthetic

data. But compared to this study, they used less scenario profile shapes (Bösch et al. 2018) or they restricted their investigations to a set of profiles with fixed combinations of shapes and vertically integrated quantities (VCDs and AOD). Most importantly, in this study, we cover a larger range of VCDs and AODs, including especially high values

5    (AODs up to 5, and $NO_2$ VCDs up to $10^{16}$ molecules cm$^{-2}$), while previous studies used maximum $NO_2$ VCDs of $2\times10^{16}$ molecules cm$^{-2}$ and $3.5\times10^{16}$ molecules cm$^{-2}$, respectively and maximum AODs of 1. Also our study investigates the trace gas retrievals for a minimum $NO_2$ VCD of $0.1\times10^{16}$ molecules cm$^{-2}$. Using these wide ranges of VCDs and AODs revealed new effects and/or confirmed earlier findings in

10   more detail. The most important findings are:

(1) With increasing AOD the retrieved AODs systematically underestimate the true AODs. The underestimation reaches values of >40% and >50% for AODs of 3 and 5, respectively. The largest underestimation is found for Gaussian profiles, while for exponential profiles with scale height of 0.5 km the smallest

15       underestimation is found. These results confirm results from previous studies with similar findings (e.g. Irie et al., 2008; Bösch et al., 2018; Frieß et al., 2019; Tirpitz et al., 2021). However, in this study, the range of AODs and the variety of profile shapes is much larger, which allows a more detailed interpretation of the results. Interestingly, the underestimation is systematically smaller for MAPA compared

20       to PriAM, which indicates that only a part of the underestimation can be attributed to the missing sensitivity of MAX-DOAS measurements towards higher altitudes.

In most cases, the larger effect for OE algorithms is probably due to the smoothing effect.

(2) Another important finding of this study is that the $NO_2$ profiles are not very sensitive to the aerosol profiles confirming similar findings by Frieß et al. (2019).

(3) Further, it was found that the influence of the assumed asymmetry parameter and single scattering albedo have typically a minor effect on the retrieval results. This is an important result, because usually the optical properties of aerosols are not well known. However, for aerosol inversions, the errors can still be up to 25%. Thus it is still important to use reasonable values for both parameters to minimize the remaining uncertainties. For the $NO_2$ inversion the influence of the asymmetry parameter and single scattering albedo is smaller, similar as found by Hong et al. (2017).

(4) Another important finding of this study is that the $NO_2$ VCDs either systematically overestimate (for low $NO_2$ VCDs) or underestimate (for high $NO_2$ VCDs) the true $NO_2$ VCDs. Interestingly, these results are rather insensitive to the shape or the AOD of the respective aerosol profiles. The underestimation for high $NO_2$ VCDs is a new finding which was not reported so far. It is probably caused by non-linearities in the radiative transport for strong $NO_2$ absorptions. It can reach deviations of more than $-30\%$ for a $NO_2$ VCD of $10^{16}$ molecules $cm^{-2}$. A tendency of an overestimation for small $NO_2$ VCDs was already observed (for OE algorithms) by Frieß et al. (2019), but not discussed in detail. Our results clearly indicate that the overestimation systematically increases towards small $NO_2$ VCDs

(with deviations >50% for an NO$_2$ VCD of $0.1 \times 10^{16}$ molecules cm$^{-2}$). Here it is interesting to note that similar results are found for different profile shapes. This finding is probably caused by the fact that the trace gas VCD is mostly constrained by measurements at high elevation angles and the fact that the trace gas SCDs for these elevation angles only weakly depend on the profile shape. Overall, the reason for the underestimation of the retrieved NO$_2$ VCD for low NO$_2$ VCDs is not yet fully understood. However, for the OE algorithm it might be caused by the influence of the a priori profile on the retrieval result. Interestingly, in this study a similar underestimation was also found for the parameterised algorithm (which was not observed by Frieß et al., 2019). This finding is currently unexplained, but might be caused by the different radiative transfer models used for the generation of the synthetic data (SCIATRAN) and in the MAPA inversion algorithm (MCARTIM). This aspect should be further investigated in future studies. Interestingly, an overestimation of the true NO$_2$ VCDs (derived from direct sun observations) by the retrieved NO$_2$ VCDs from MAX-DOAS observations was also reported by Tirpitz et al. (2021) for low NO$_2$ VCDs (but not for HCHO VCDs).

(5) Another important finding of our investigations confirms the results from earlier studies (e.g. Wang et al., 2017; Bösch et al., 2018). Changing the covariance matrix changes also the retrieval results from OE retrieval as it results in different weighting of a priori and measurements in the inversion.

**4 Conclusions**

Given that severe air pollution often occurs during autumn and winter in China, the effects of different aerosol conditions on the accuracy of MAX-DOAS profile retrieval were studied. The effects of aerosols on MAX-DOAS retrievals of aerosols and $NO_2$ profiles were examined by assuming a series of aerosol scenarios with 3 aerosol profile

shapes (exponential, Boltzmann, and Gaussian) with AODs/VCDs ranging from 0.1 to 5.0 at two wavelengths (360nm and 477nm). In addition, a series of $NO_2$ scenarios was assumed with the same profile shapes and various VCD values (from $0.1 \times 10^{16}$ to $10.0 \times 10^{16}$ molecules $cm^{-2}$). Compared to previous studies (e.g. Bösch et al., 2018; Frieß et al., 2019) our input profiles cover a much larger range of AODs and $NO_2$ VCDs and

also more profile shapes and more combinations between them.

In a first step, the effects of the assumed single-scattering albedo (SSA) and asymmetric parameter (AP) on the aerosol profile inversion was investigated. It was found that the retrieved aerosol extinction profiles are very consistent if the same SSA and AP values are used for the simulations of the $O_4$ DSCDs and the PriAM inversions. If incorrect

SSA and AP values were used, the retrieved extinction coefficients were smaller than the input values in the case of too low of AP or too high SSA assumed in the profile inversion and vice versa (with opposite behavior for the surface values). However, for most cases the deviations caused by wrongly assumed AP and SSA were found to be rather small compared to other uncertainties. The maximum relative deviation was

generally found around 1.0km with the values of about 25%.

Next, the differences of the PriAM and MAPA profile retrievals from the input profiles for different aerosol conditions were examined. We found that both algorithms have

systematic deficiencies in retrieving the 4 profile shapes. Especially at low (above 0.2 km) and high (above 1.5 km) altitudes, often deviations from the true values are found, while for altitudes in between best agreement is found. The algorithms can reasonably retrieve the 4 aerosol profile shapes of AODs < 1.0 for two wavelengths, but for AODs > 1.0 the retrieved values systematically underestimate the true AODs. The smallest magnitude of the relative deviations (typically <20%) were found for exponential profile shapes, with a scale height of 0.5 km. Large magnitude of the relative deviations (up to >50%) are found for the other profile shapes, especially for high AODs. Such a systematic underestimation has also been found in several previous studies (e.g. Irie et al., 2008, Frieß et al., 2016, Bösch et al. 2018, and Tirpitz et al., 2021). The systematic deviation between MAX-DOAS and sun photometers is partly caused by the missing sensitivity of MAX-DOAS observations for higher altitudes and the smoothing effect, especially for optimal estimation algorithms (e.g. Tirpitz ez al., 2021). In general, the relative deviations of the MAPA results depend less on the AOD than the PriAM results. For MAPA, part of the differences between input and retrieved AODs can be explained by the differences in the RTM model. It should also be noted that for the Gaussian profiles, both PriAM and MAPA could retrieve the lifted layer. However, PriAM underestimated the width of the lifted layer and the extinction coefficient at the peak, while MAPA overestimated the width of the lifted layer and significantly underestimated the aerosol extinctions at the peak.

Then, for PriAM, the effect of using different *a priori* profiles and *a priori* profile covariance matrices (Sa) was studied. The results showed that the retrieval results of

the aerosol profiles were slightly improved when the same *a priori* profile shape as the

input profile shape was used. The main reason is probably that the corresponding *a*

*priori* bias was reduced. In addition, the inversion results were more consistent with the

input profiles when the AOD of the *a priori* profile was increased for high AOD

scenarios. The effect of the Sa value for the 4 aerosol shapes was investigated for the

extreme scenario with an AOD of 5.0. It was found that the correlation coefficient could

be improved by increasing the Sa values for all aerosol profile shapes, mainly because

of improved values of the retrieved surface extinction and scale height.

Also the modeled $O_4$ DSCDs corresponding to the aerosol profiles retrieved by PriAM

and MAPA were compared to $O_4$ DSCDs simulated by the RTM for the input aerosol

profiles. The averaged correlation coefficients of the modeled and simulated $O_4$ DSCDs

were > 0.99 for both PriAM and MAPA, indicating that a possible non-convergence of

the profile retrievals is not a reason for the systematic discrepancies of retrieved profiles

from the input profiles.

In the next part, the effects of the aerosol retrieval on the $NO_2$ profile retrieval were

studied for PriAM and MAPA. Two strategies were utilized to retrieve the $NO_2$ profiles,

in which either the retrieved or the input aerosol profiles served as input for the

retrievals of the $NO_2$ profiles in strategy 1 (S1) and strategy 2 (S2), respectively.

Strategy S1 was applied both to PriAM and MAPA, while strategy S2 was only applied

to PriAM.

From these studies several conclusions could be drawn: The relative deviations of the

retrieved $NO_2$ VCDs do only slightly depend on the AOD or the shape of the aerosol

profiles. In contrast, especially for PriAM, a systematic dependence on the $NO_2$ VCD was found. For low $NO_2$ VCDs the retrieved $NO_2$ VCDs largely underestimate the true $NO_2$ VCDs by up to 60%, while for high $NO_2$ VCDs a systematic underestimation up to -30% is found. Here it should be noted that in spite of the large relative deviations

for low $NO_2$ VCDs, the absolute deviations are rather small. The underestimation of the true $NO_2$ VCD for high $NO_2$ VCDs by the retrieved profiles was not reported before. It is probably caused by non-linearities in the radiative transport for strong $NO_2$ absorptions. The increase of the Sa values did not improve the inversion results for high AODs, but instead lead to the occurrence of single outliers in some layers.

We also performed a consistency check of the optimal estimation algorithm by using exactly the a priori profiles as input profiles. For the aerosol retrieval, almost the exact input profiles were retrieved (differences < 0.05%) indicating that there are no inconsistencies in the algorithm. However, for the trace gas profiles no such perfect agreement was found, especially towards scenarios with high AODs and $NO_2$ VCDs

indicating the more complex dependencies of trace gas retrievals compared to aerosol retrievals. Here it is important to note that the relative deviations for the retrieved $NO_2$ profile by using both the aerosol and $NO_2$ *a priori* profiles as input profiles are smaller than those for scenarios for which only the aerosol a priori profile is used as input profile. Finally it should be mentioned that the results of this study are very similar for both

selected wavelengths (360 and 477 nm) indicating that the differences in the radiative transfer between both wavelengths have no strong effect on the MAX-DOAS profile retrievals.

**Author contributions**

XT, YW, SB, PX and TW contributed to design the research. SB performed the MAPA profile inversions and calculated the input profiles. SD convert $O_4$ data format to MAPA input data format. BR and XL processed the SCIATRAN data. XT performed the data analyses and wrote the manuscript. JX and AL supervised this study and provided suggestions for the manuscript. PX and YW revised this manuscript. TW and SB developed the manuscript.

**Acknowledgment**

This study was supported by the National Natural Science Foundation of China (Grant Nos.: 41530644, U19A2044, 41975037), Natural Science Foundation of Anhui Province (Grant Nos.: 2008085QD183, 2008085QD182), and the Open Fund of Key Laboratory of Environmental Optics and Technology, Chinese Academy of Sciences (Grant Nos.: 2005DP173065-2019-04).

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

**Table 1. Parameter settings used in the RTM. Default values are indicated by [*].**

| Parameters | | |
|---|---|---|
| Target species | aerosol, $NO_2$ | |
| Wavelength (nm) | 360, 477 | |
| Single scattering albedo (SSA) | 0.8, 0.9[*], 1.0 | |
| Asymmetry parameter (AP) | 0.65, 0.72[*] | |
| Solar zenith angle(SZA, °) | 20, 40, 60, 80 | |
| Relative azimuth angle (RAA, °) | 30, 60, 120, 180 | |
| Elevation angles (EA, °) | 1, 2, 3, 4, 5, 6, 8, 15, 30, 90 | |
| Aerosol optical depth (AOD) | 0.1, 0.3, 1.0, 3.0, 5.0 | |
| $NO_2$ Vertical column density (VCD, $10^{16}$ molecules $cm^{-2}$) | 0.1, 0.3, 1.0, 3.0, 10.0 | |
| Profile types and parameters | Exponential: | scale heights 0.2, 0.5[*], 1.0[*] km |
| | Gaussian: | peak heights 0.5, 1[*] km; |
| | | peak widths 0.2, 0.5[*], 1.0, 1.5 km |
| | Boltzmann: | heights 1.0, 1.5[*], 2.0 km |

**Table 2. List of retrieval algorithms used in the comparison**

| Algorithm | Forward Model | Method |
|---|---|---|
| PriAM | SCIATRAN version 2.2 | OEM (Optimal Estimation Method) |
| MAPA | McArtim | Parameterized retrieval in combination with Monte Carlo approach |

**Table 3. List of SSA and AP values used for the sensitivity studies (for the standard retrievals SSA = 0.9 and AP = 0.72 were used).**

| Parameters | |
|---|---|
| SSA | 0.7, 0.8, 1.0 |
| AP | 0.65, 0.68, 0.76, 0.80 |

**Table 4. Parameter settings used in the general PriAM retrieval for aerosol and NO₂ profiles.**

| Parameters | | |
|---|---|---|
| a priori profile | Aerosol : | exponential shape with an AOD of 0.2 and the scale height of 1.0km |
| | NO$_2$: | exponential shape with the VCD of $1.0 \times 10^{15}$ molecules cm$^{-2}$ and the scale height of 1.0km |
| Sa_ratio | Aerosol : | 0.1 |
| | NO$_2$: | 0.5 |

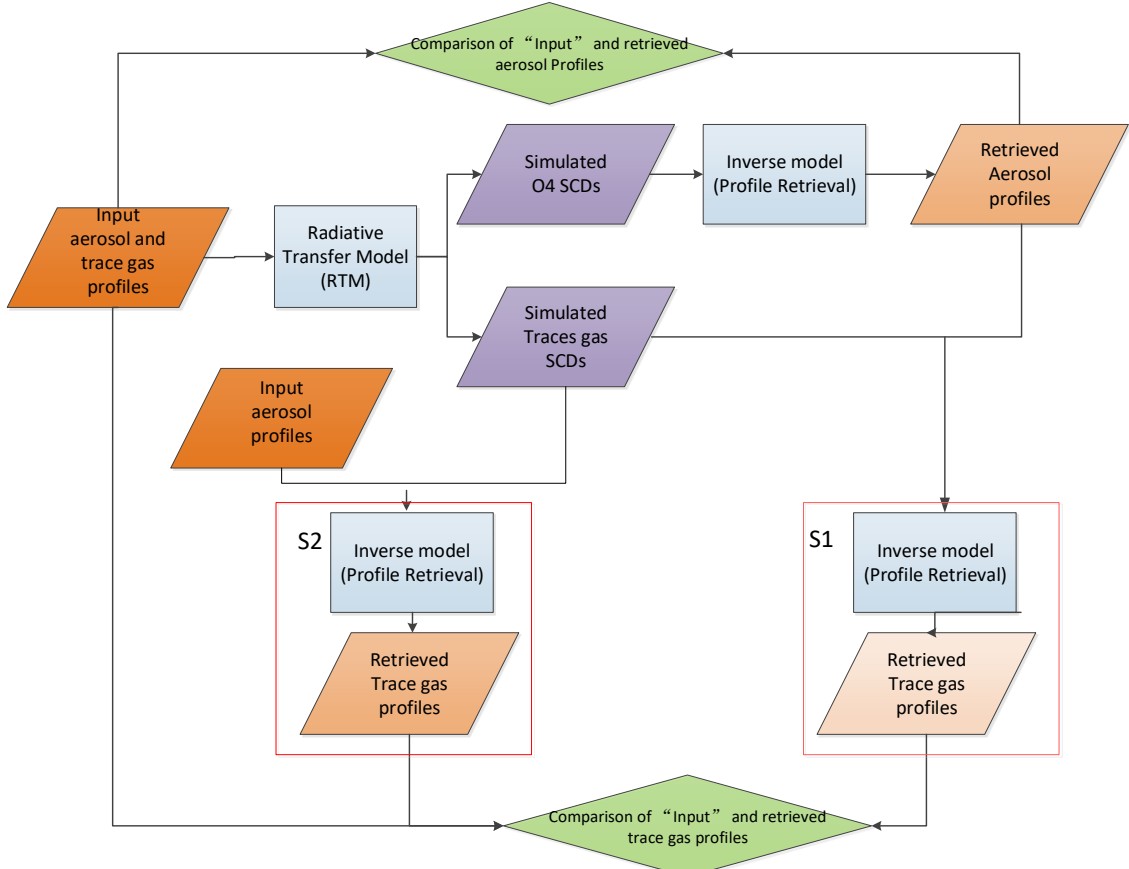

**Figure 1. Flow diagram depicting the strategy used for the analysis of the effects of high aerosol loads on the retrieval of aerosol and trace gas profile**

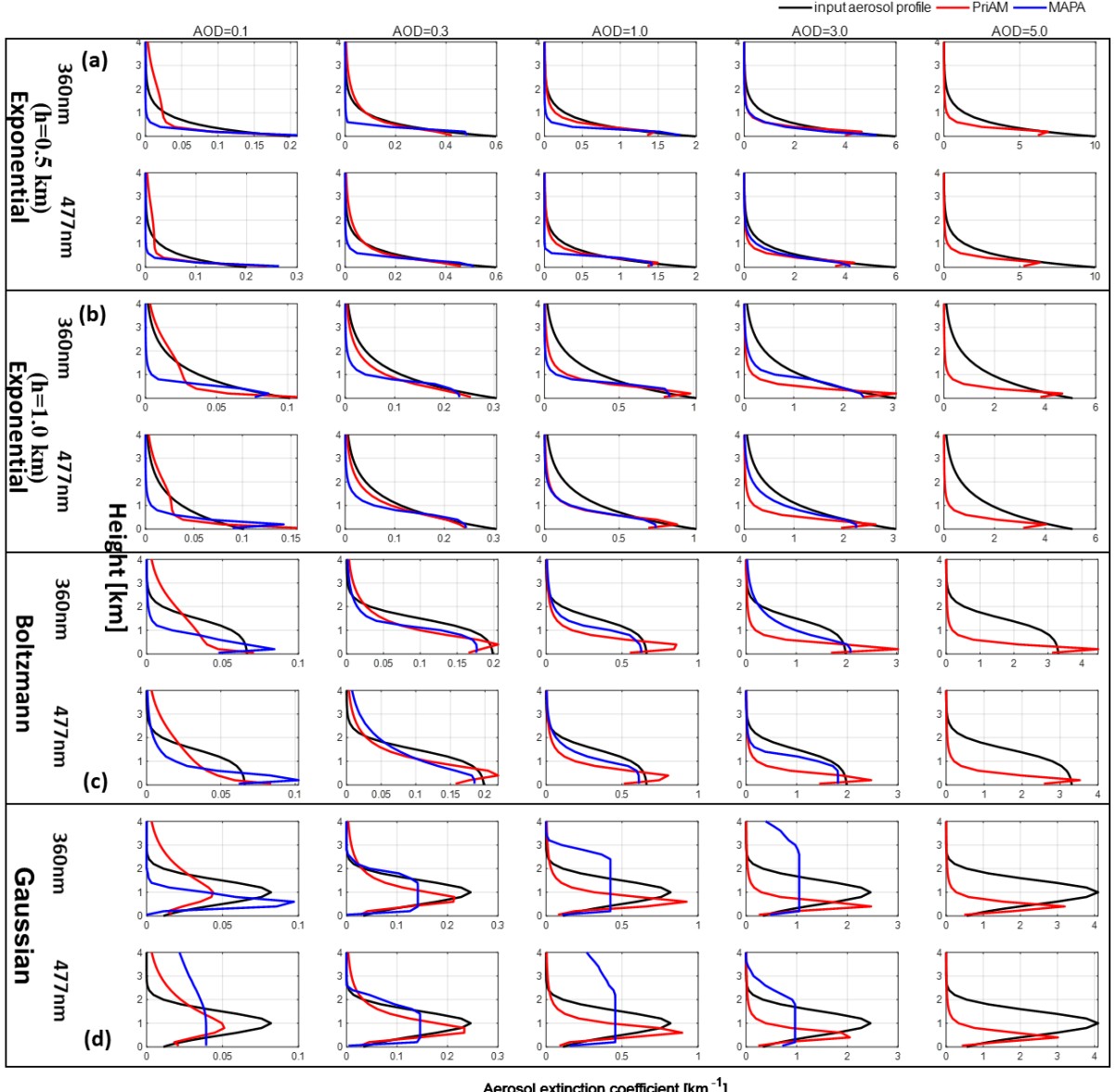

**Figure 2. Comparison of the aerosol profiles retrieved by PriAM and MAPA for 360 nm (first line) and 477 nm (second line) and the corresponding input aerosol profiles for (a) exponential shape with h = 0.5 km, (b) exponential shape with h= 1.0 km, (c) Boltzmann shape, and (d) Gaussian shape.**

The red and blue curves indicate the results from PriAM and MAPA, respectively. The corresponding relative deviations and absolute deviations are shown in Fig. S8 and Fig. S9, respectively. Note that MAPA by default flags cases where the retrieved AOD exceeds 3, thus the high aerosol scenarios are missing for MAPA.

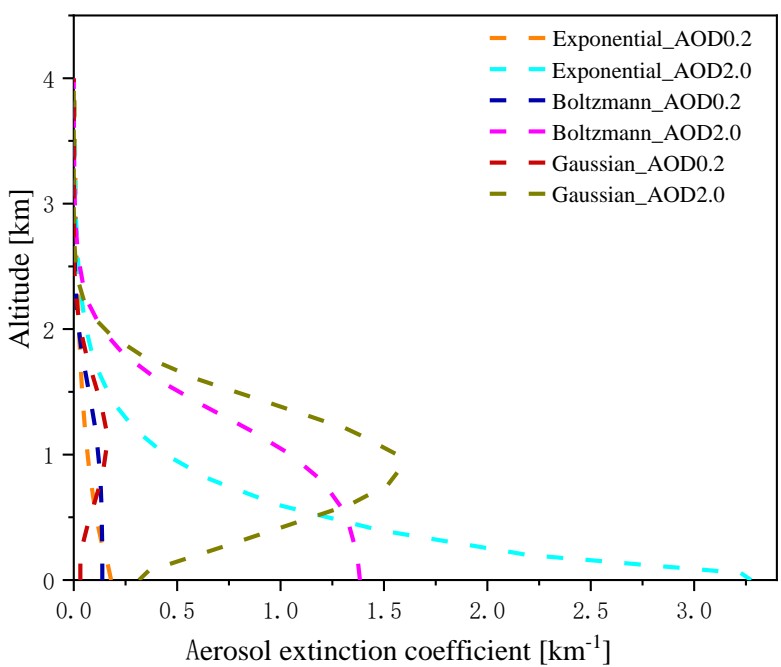

**Figure 3. Different aerosol *a priori* profiles used by PriAM in this study.**

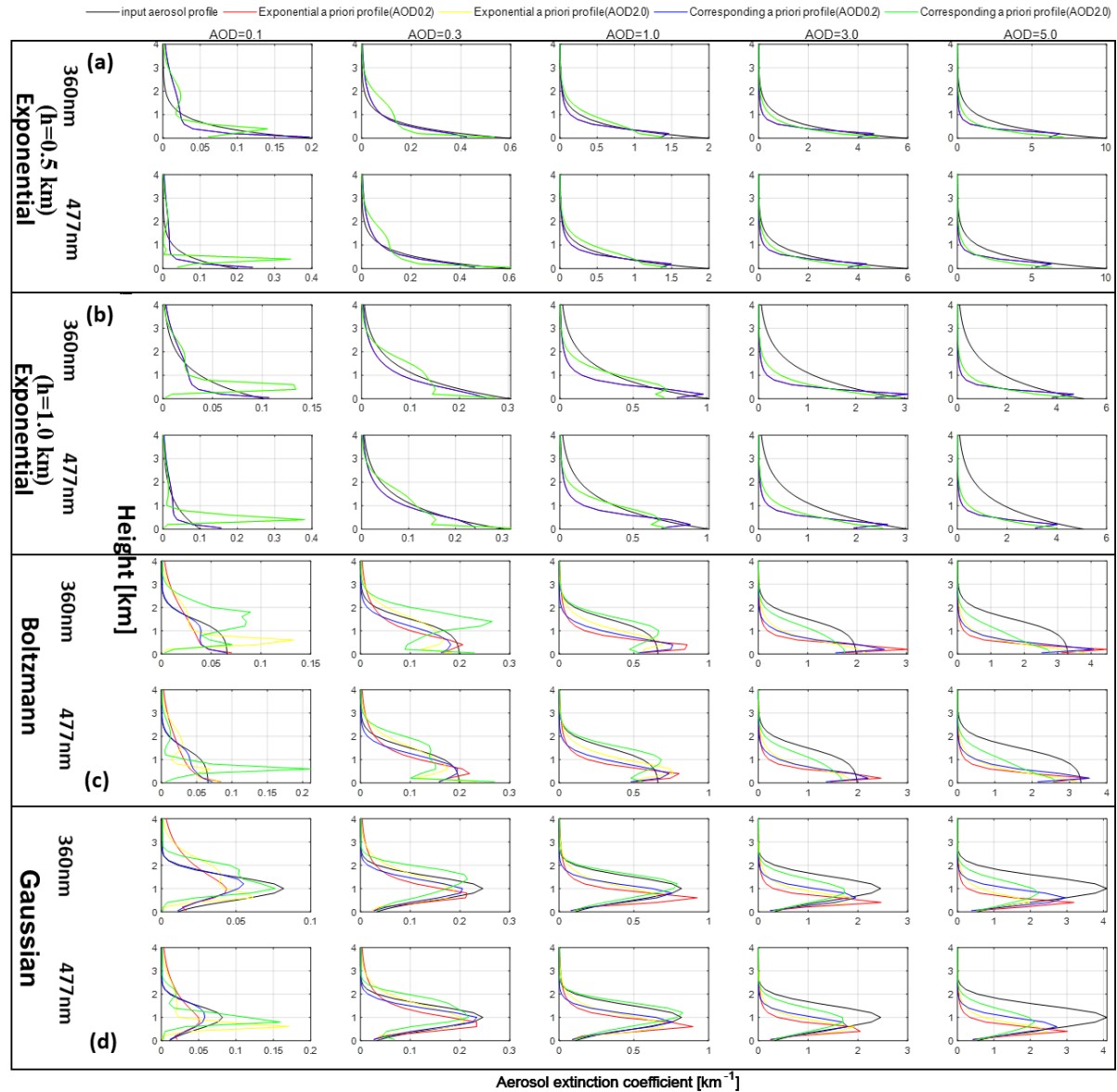

**Figure 4. Comparison of the PriAM inversion results with different alternative a priori profiles and the results for the universal *a priori* (exponential shape with AOD 0.2).**

The first line in every panel denotes the results for 360 nm, and the second line denotes the results for 477 nm. Colors indicate the shapes and AODs shown at the top. 'Corresponding *a priori* profile' means that the same profile type as the simulated profiles is also used as *a priori* profile.

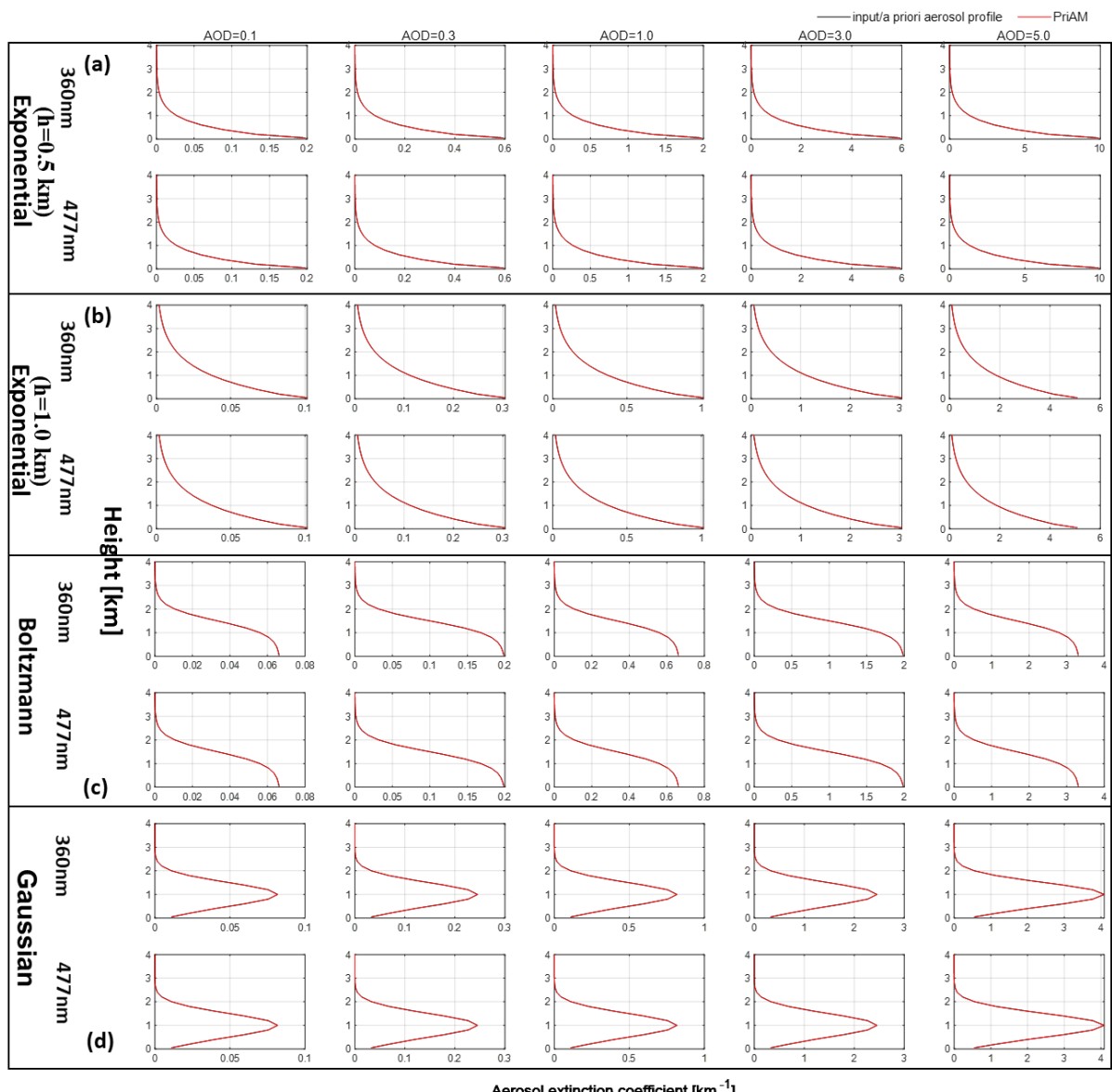

**Figure 5. Comparison of the input aerosol profiles and the PriAM inversion results if the exact a priori profiles are used as input profiles.**

The first line in every panel denotes the results for 360 nm, and the second line denotes the results for 477 nm. The black and red curves indicate the input (the same as the *a priori* profile) and retrieved aerosol profiles by PriAM, respectively.

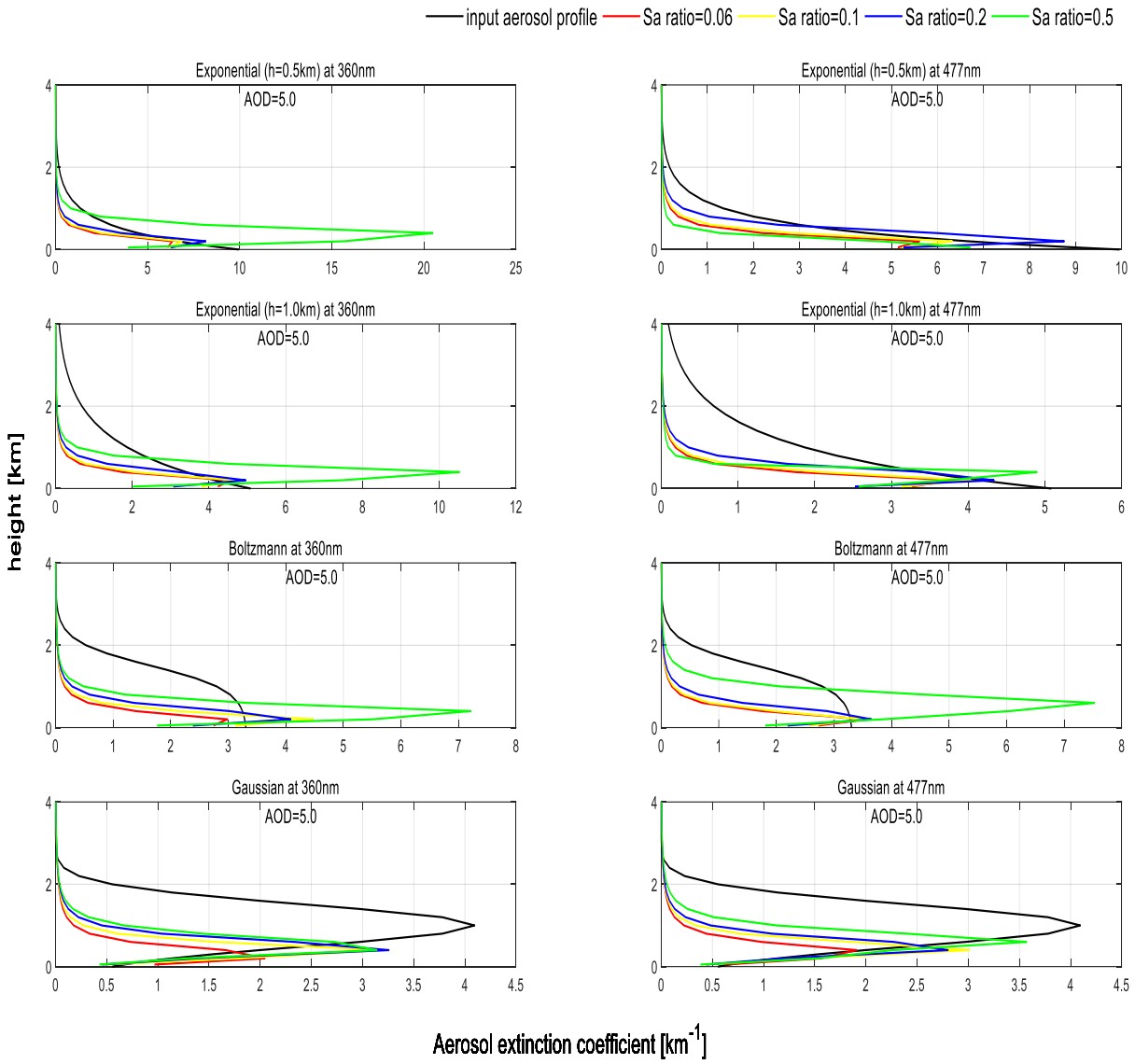

**Figure 6. Results for three aerosol profile shapes retrieved by PriAM for an AOD of 5.0 by using different values of the *a priori* profile covariance matrix (Sa).**

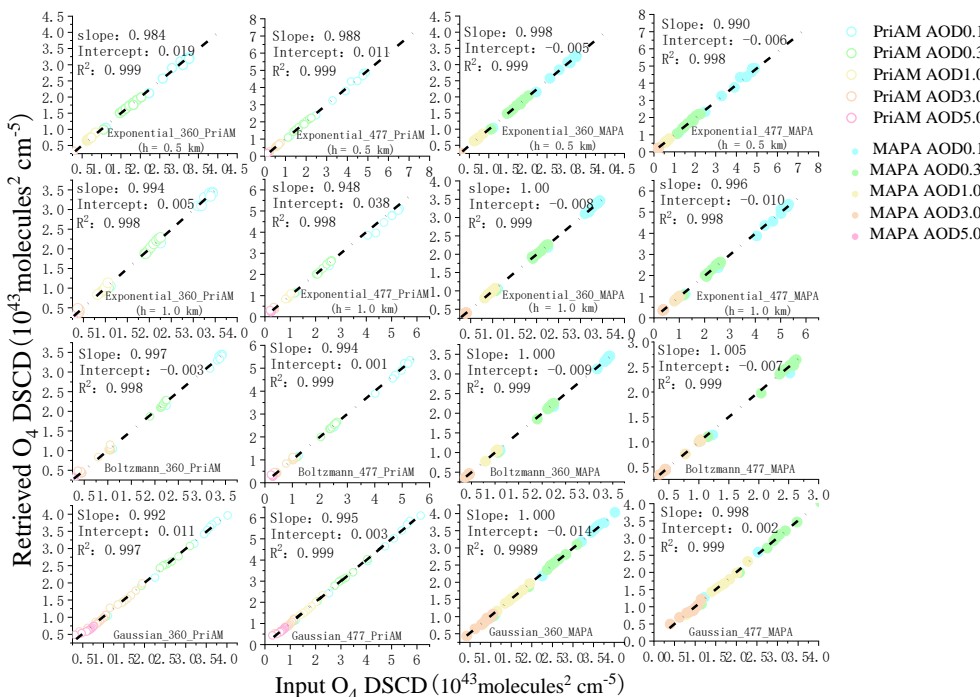

**Figure 7. Correlation plots between the retrieved O₄ DSCDs and the input O₄ DSCDs for PriAM and MAPA**

The open and closed circles denote the retrieved O₄ DSCDs from PriAM and MAPA, respectively.

The colors refer to the AOD shown at the top right.

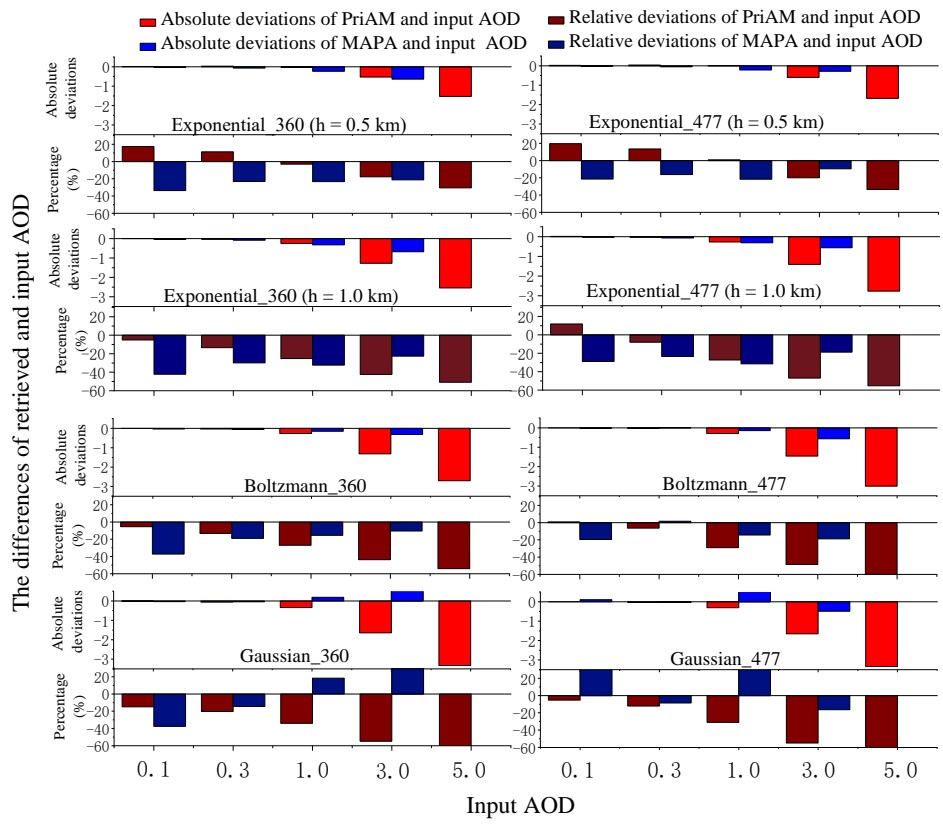

**Figure 8. Comparison between the retrieved AODs and input AODs for PriAM and MAPA for the 4 aerosol profile shapes listed in table 1.**

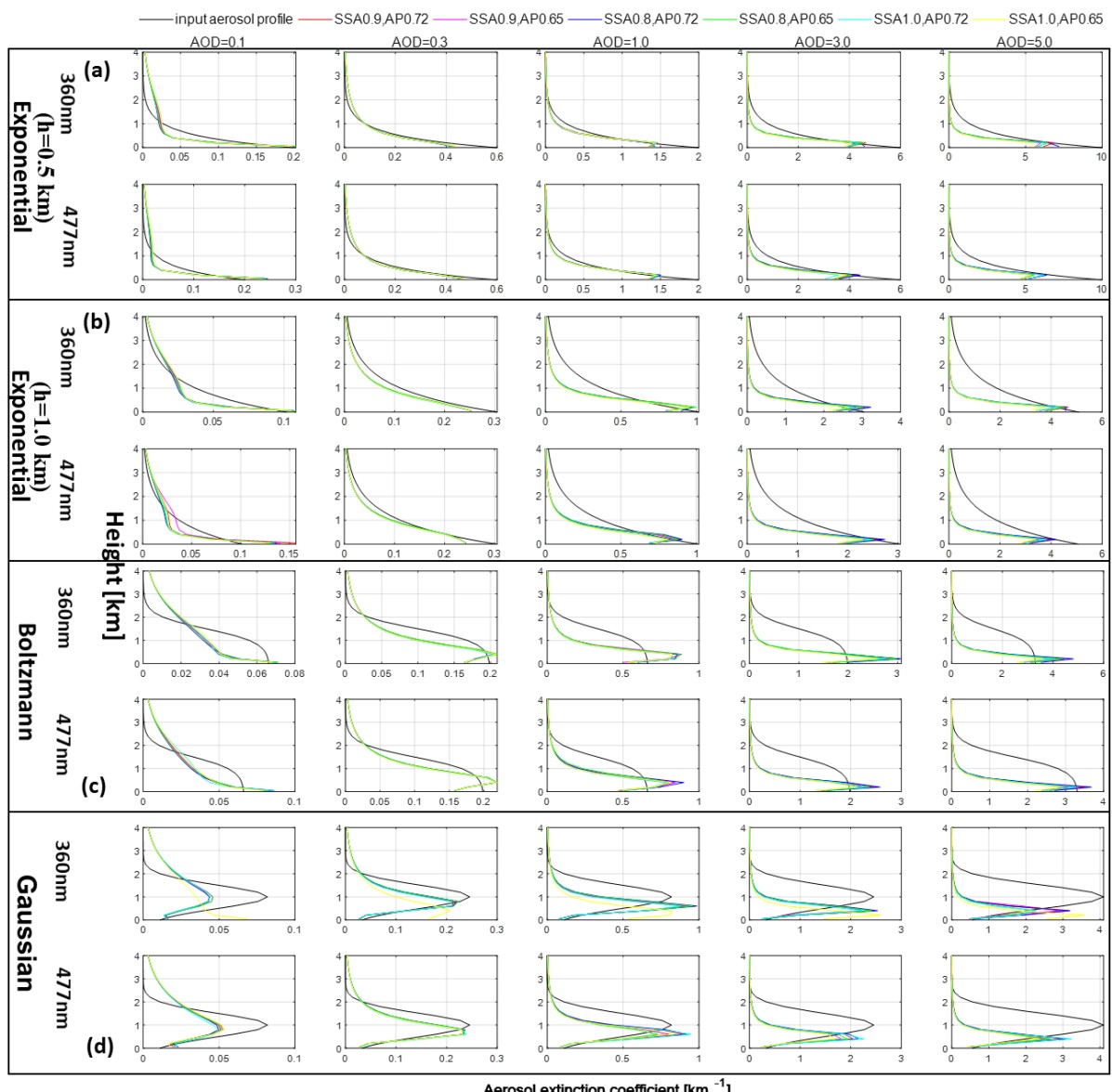

**Figure 9. Retrieved aerosol profiles by PriAM using different SSA and AP (a) exponential shape with h = 0.5 km, (b) exponential shape with h = 1.0 km, (c) Boltzmann shape, and (d) Gaussian shape. For these inversions, the same SSA and AP were used for the simulations of the O₄ DSCDs and for the PriAM inversions. The first line in every panel denotes the results for 360 nm, and the second line denotes the results for 477 nm.**

**The colors refer to the corresponding SSA and AP values shown at the top.**

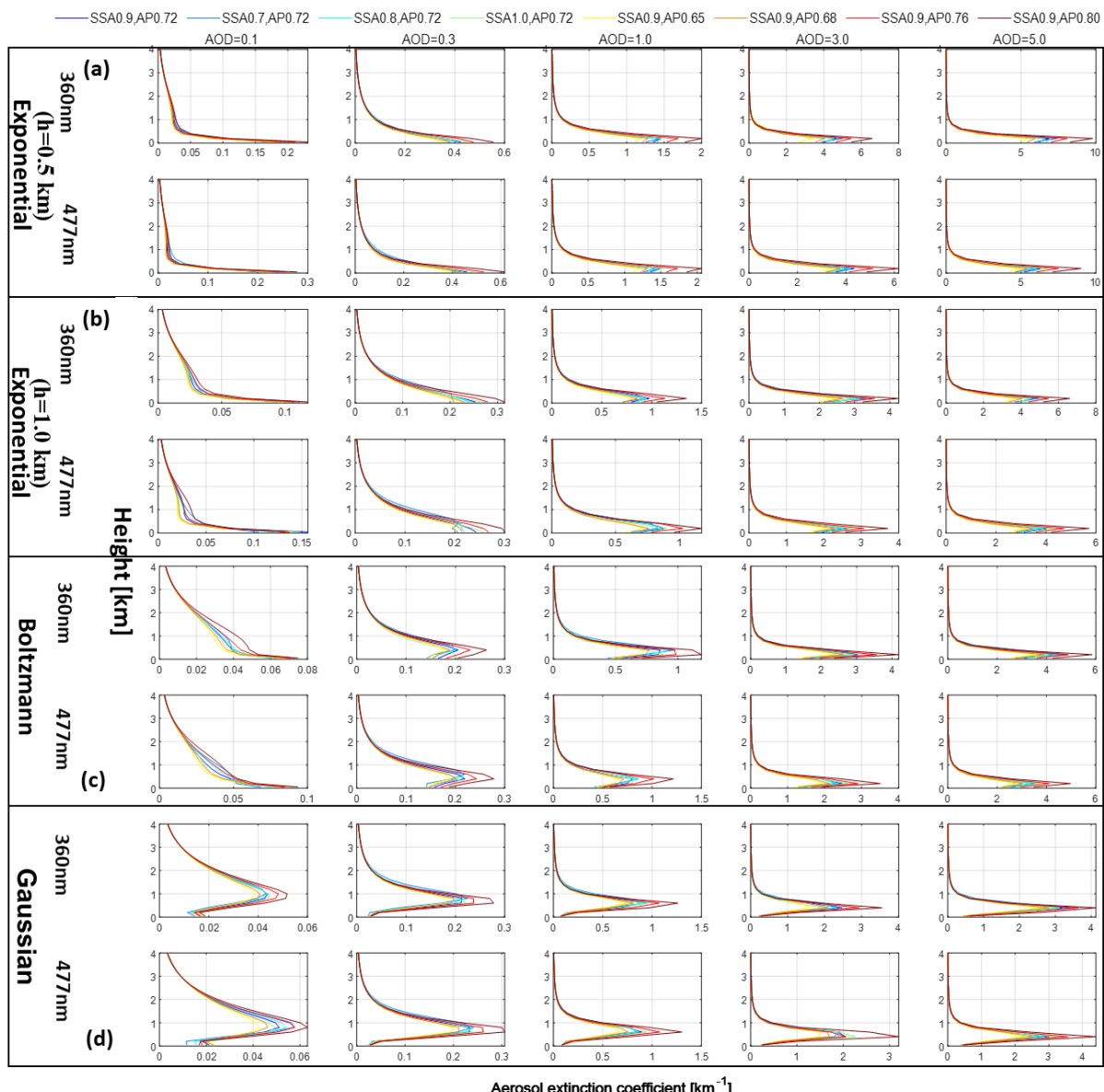

**Figure 10. The retrieved profiles using incorrect SSA and AP values from the retrieved profiles with the correct SSA and AP values for (a) exponential shape with h = 0.5 km, (b) exponential shape with h = 1.0 km, (c) Boltzmann shape, and (d) Gaussian shape.**

The colors refer to the SSA and AP values shown at the top.

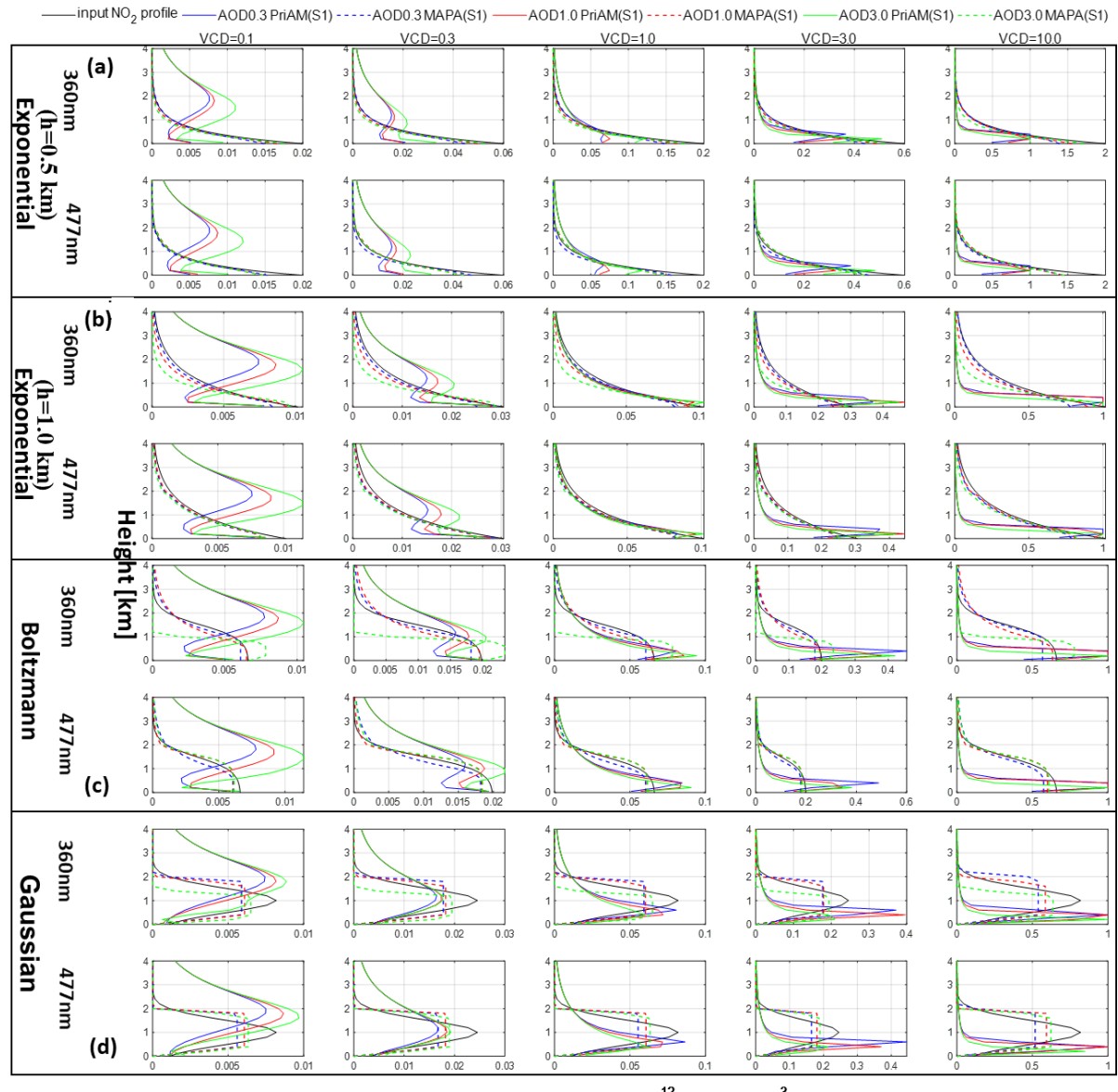

**Figure 11.** Retrieved NO₂ profiles by PriAM and MAPA for scenario S1 (see text) for aerosol profiles with 3 selected AODs (0.3, 1.0, and 3.0) and 5 NO₂ VCDs for of (a) exponential shape with h = 0.5 km, (b) exponential shape with h = 1.0 km, (c) Boltzmann shape, and (d) Gaussian shape.

The solid and dotted colored lines refer to the AODs and algorithms shown at the top

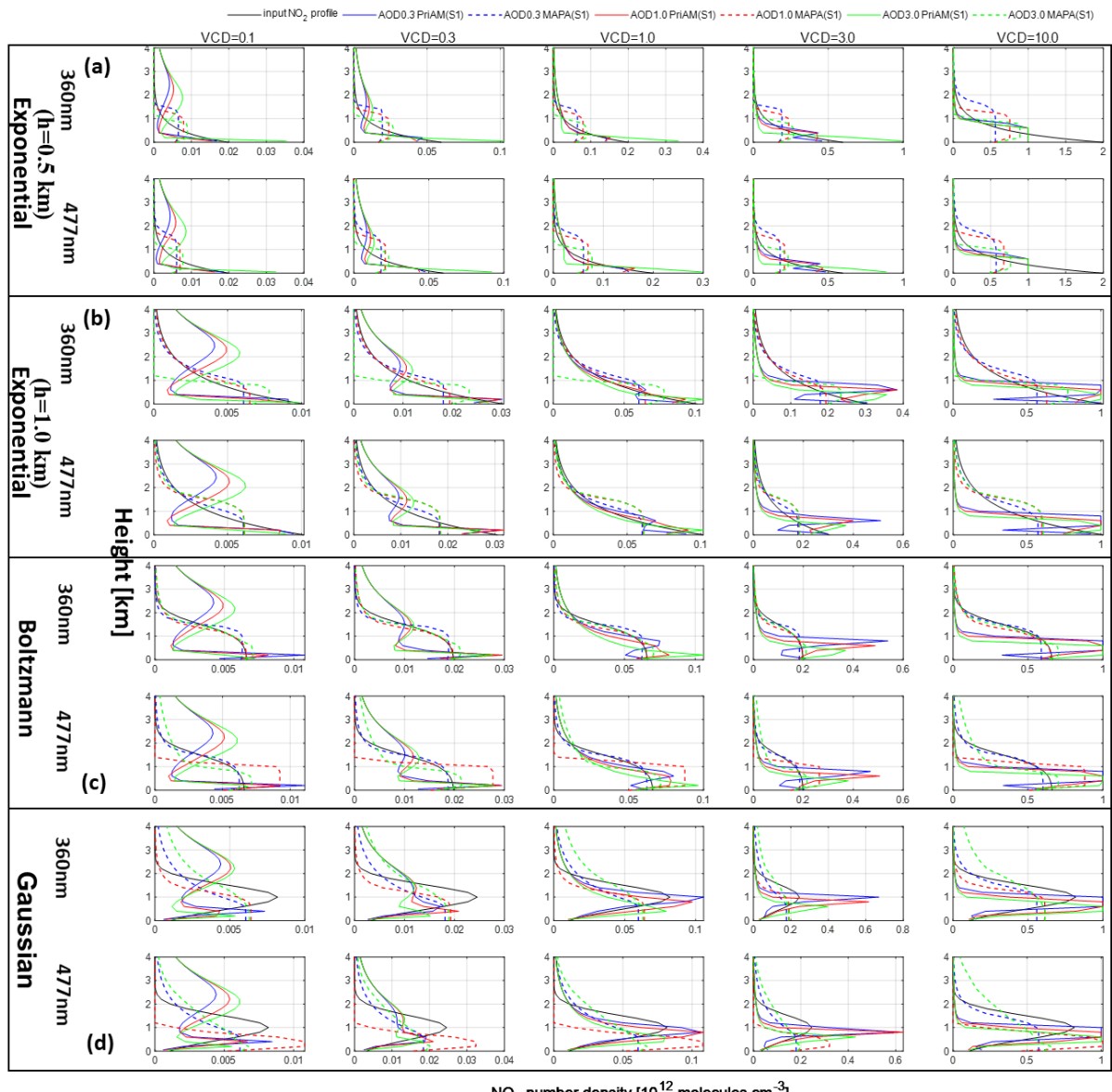

**Figure 12. Retrieved NO₂ profiles by PriAM and MAPA for Boltzman NO₂ input profiles for scenario S1 (see text) and for 3 aerosol profile shapes ((a) exponential shape with h = 0.5 km, (b) exponential shape with h = 1.0 km, (c) Boltzmann shape, and (d) Gaussian shape) with 3 selected AODs (0.3, 1.0, and 3.0) and 5 NO₂ VCDs.**

**The solid and dotted colored lines refer to the AODs and algorithms shown at the bottom right.**

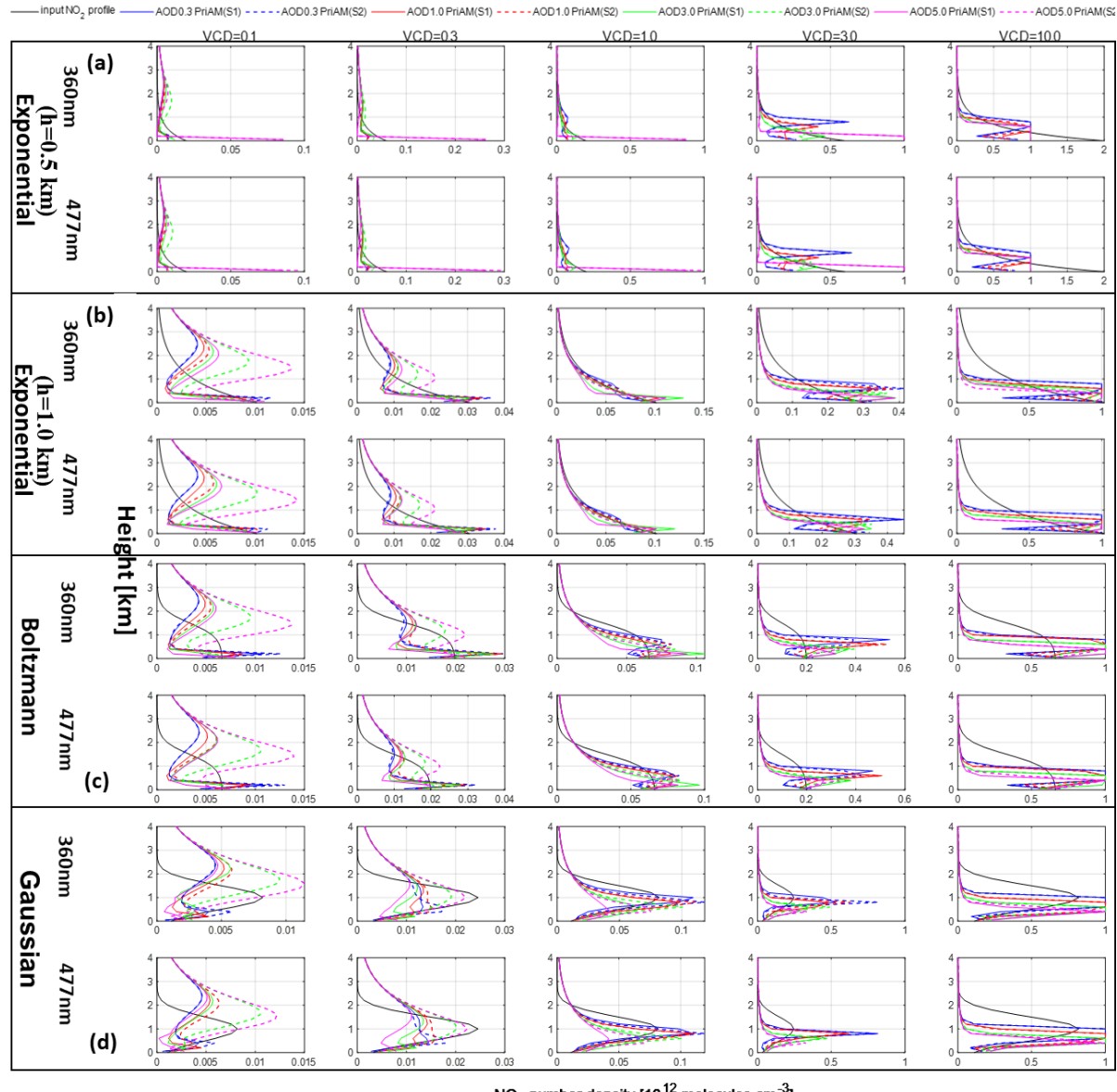

**Figure 13. Retrieved NO₂ profiles by PriAM for scenarios S1 and S2 and input NO₂ profiles for 4 AODs (0.3, 1.0, 3.0, and 5.0) and 5 VCDs for (a) exponential shape with h = 0.5 km, (b) exponential shape with h = 1.0 km, (c) Boltzmann shape, and (d) Gaussian shape.**

The first line in each panel denotes the results for 360 nm, and the second line denotes the results for 477 nm. The solid and dotted colored lines refer to the AODs and strategies shown at the top.

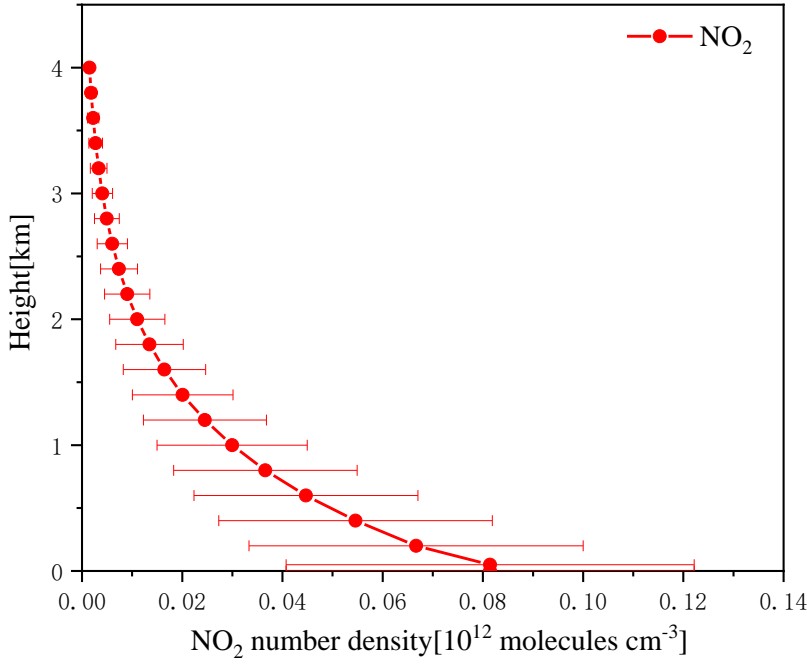

**Figure 14. The *a priori* NO₂ profiles used by PriAM for NO₂ the retrieval in this study. The error bars represent the *a priori* uncertainty.**

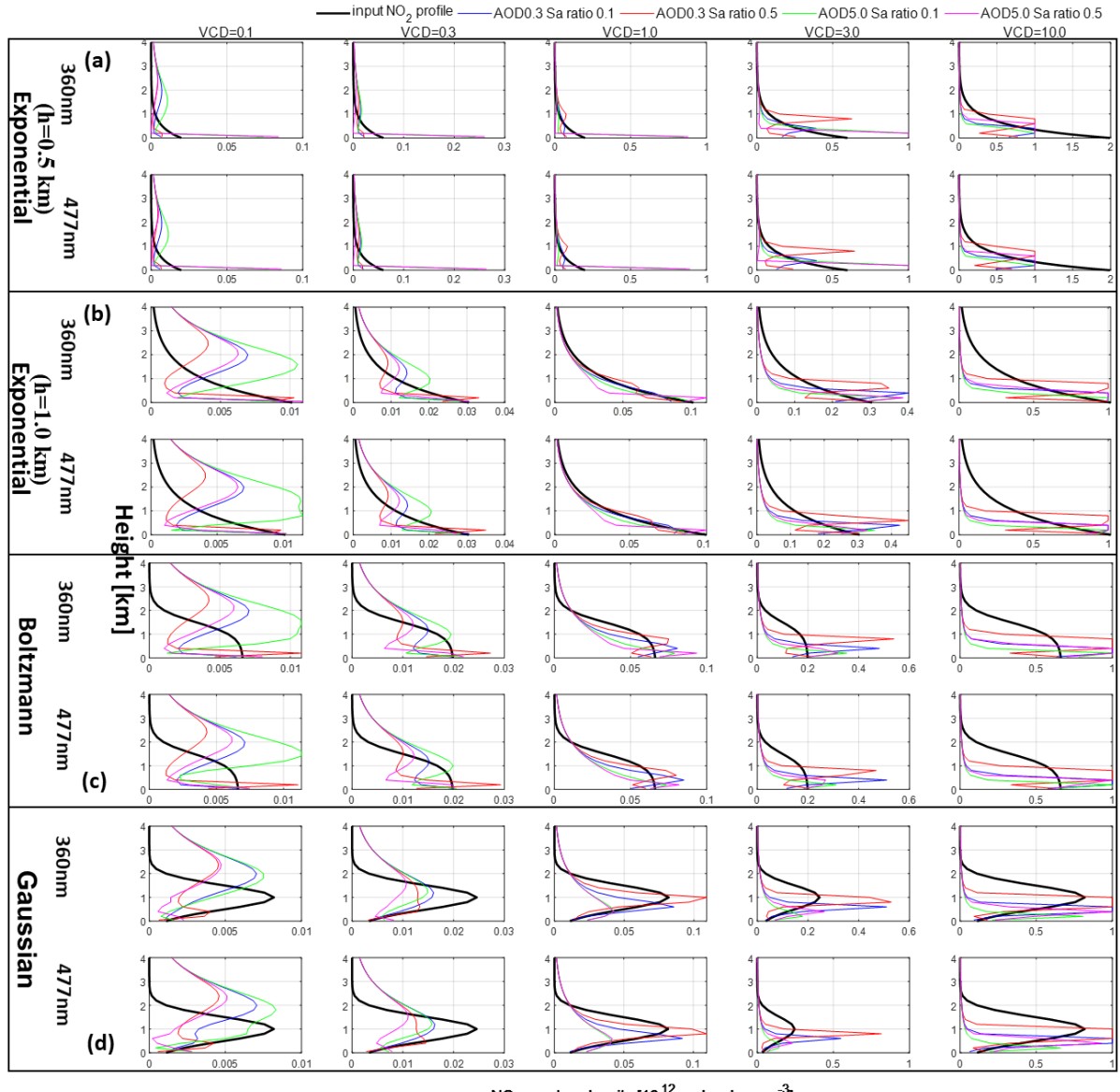

**Figure 15. Retrieved NO₂ profiles by PriAM for Sa of 0.1 and 0.5 and AOD of 0.3 and 5.0, along with the input NO₂ profiles for (a) exponential shape with h = 0.5 km, (b) exponential shape with h = 1.0 km, (c) Boltzmann shape, and (d) Gaussian shape.**

**The first line in each panel denotes the results for 360 nm, and the second line denotes the results for 477 nm. The solid and dotted colored lines refer to the AODs and Sa shown at the top.**

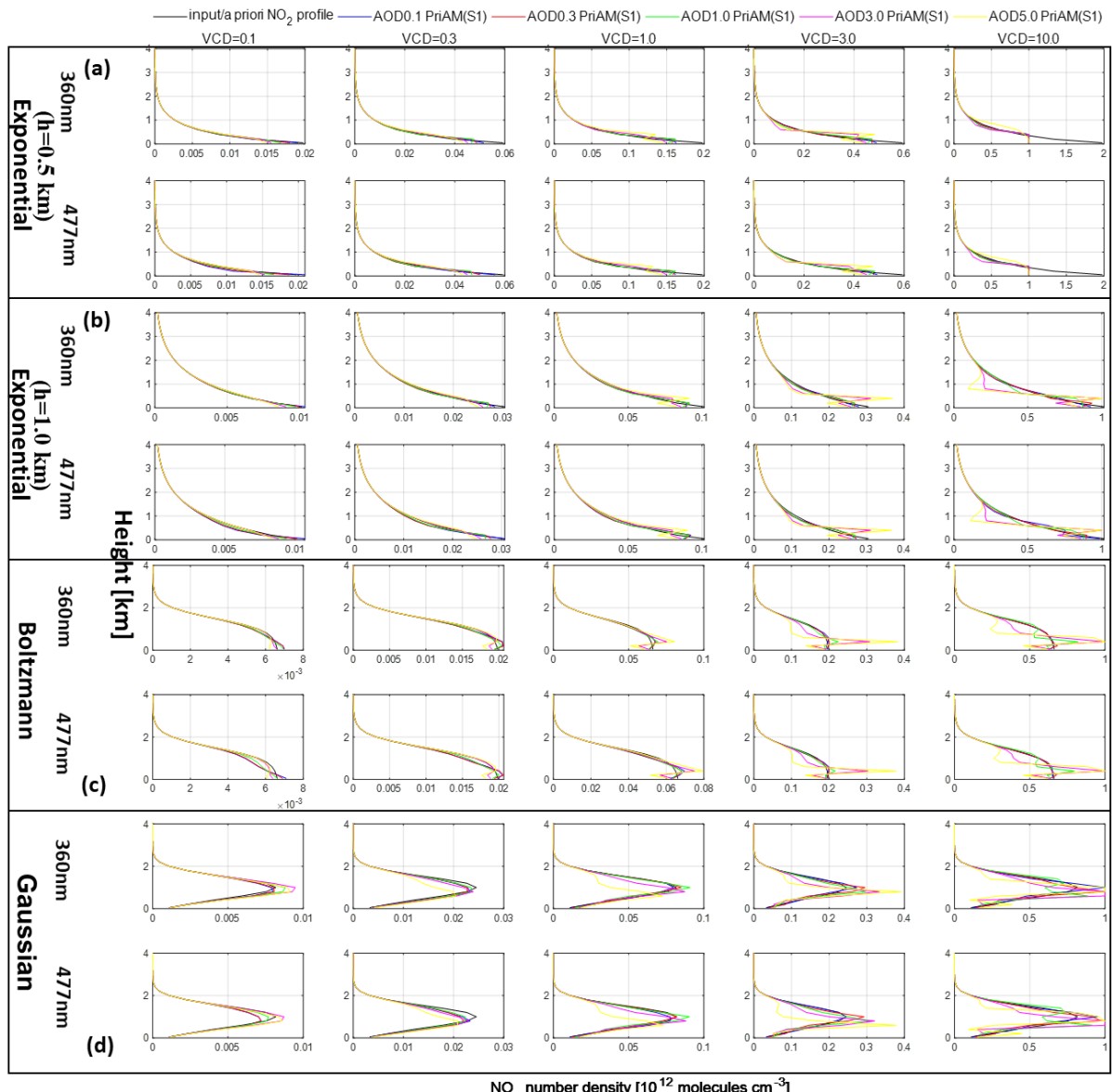

**Figure 16. Retrieved NO₂ profiles by PriAM if exactly the a priori profiles for aerosols and NO₂ are used as input profiles (for scenario S1, see text) for aerosol profiles with 5 AODs (0.1, 0.3, 1.0, 3.0, and 5.0) and 5 NO₂ VCDs for of (a) exponential shape with h = 0.5 km, (b) exponential shape with h = 1.0 km, (c) Boltzmann shape, and (d) Gaussian shape.**

**The solid colored lines refer to the AODs and algorithms shown at the top.**

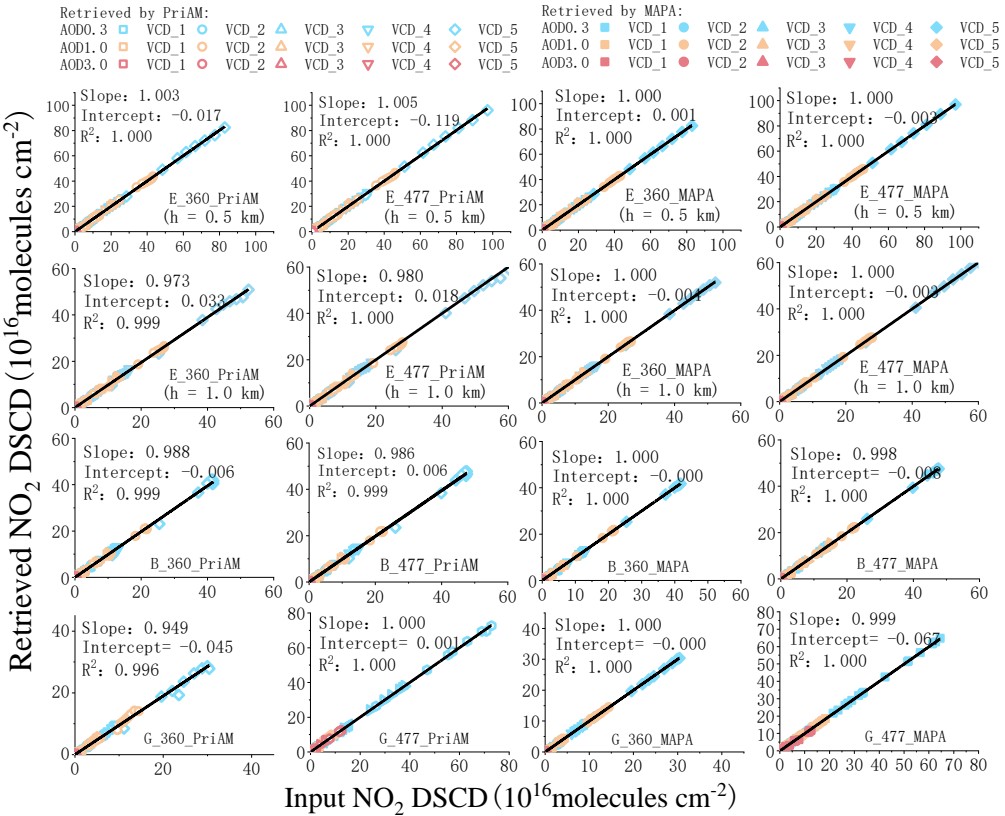

**Figure 17. Correlation plots between the retrieved NO₂ DSCDs by PriAM and MAPA versus the input NO₂ DSCDs for 3 AOD scenarios and 5 VCDs for scenario S1**

The colors refer to the VCD values and algorithms shown at the top.

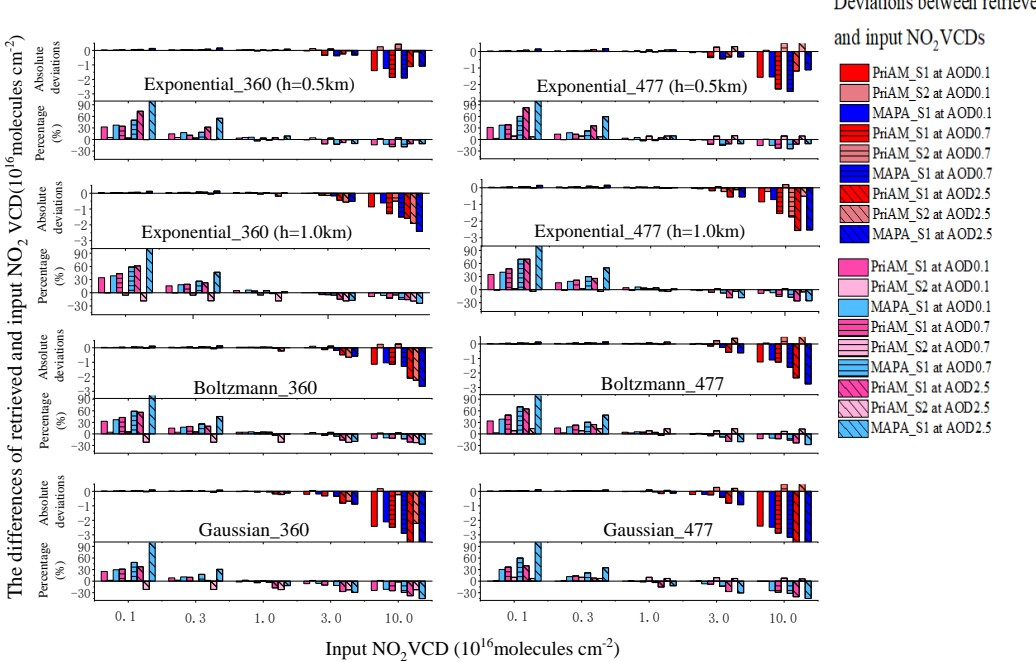

**Figure 18. Absolute and relative deviations between the retrieved and input NO₂ VCDs for PriAM (S1 and S2) and MAPA (S1) for 3 AOD scenarios and 5 VCDs.**

The colors and shapes refer to the deviations of the retrieved and input NO₂ VCDs of the different algorithms at different AODs shown at the right.