# Peer review of "Technical note: Evaluation of profile retrievals of aerosols and trace gases for MAX-DOAS measurements under different aerosol scenarios based on radiative transfer simulations"

_Atmospheric Chemistry and Physics, 2021_

## Author Comment (AC1)

Dear editors and reviewers,

Thank you very much for your constructive comments and advices on our manuscript. Your positive evaluation and comments encourage us and are great help for us. We have carefully considered every comment, and made the corresponding revisions in the revised manuscript (indicated by the 'tracked changes').

Point to point response is following:

General Comments:

1. The choice and quality of the figures can be significantly improved:

1) For many comparisons, the authors created three figures with actual profiles, absolute deviations, and relative deviations, respectively. They decided to only show the relative deviations in the main text and moved the other figures to the supplementary material. I strongly recommend showing the actually retrieved profiles in the main text and (to keep the manuscript concise) move the relative deviation to the supplementary material (e.g. swap Figure 2 and Figure S8, Figure 4 and Figure S12, and so on…). Furthermore, where possible, I recommend to also indicate the PriAM a priori profile in each subplot. Plots of this kind are easiest to read, provide very complete information (relative and absolute deviations can be readily estimated by eye), and allow to directly perceive potential impacts of a priori biases.

**Response:** Thank you very much for your suggestions. We have considered your advice to change the figures in the main text. The actually retrieved profiles were moved to the main text, and the relative deviations were moved to the supplementary material. And we also changed the corresponding content in the article.

Concerning your suggestion about including the PriAM a priori profile in each subplot, we did not follow this suggestion.

The main reason is that when we include the *a priori* profile in each subplot, the value of the a priori profile is smaller than the retrieved and true profiles. Thus we finally choose not to add the *a priori* profile in each subplot.

2) Furthermore, the vertical extent of some figures might be enhanced to improve the visibility of profile fine structures particularly close to the surface.

**Response:** Thank you very much for your suggestion. We have checked every figure and enhanced the vertical extent to make them more clear.

3) Reduce line thickness of the profiles for better visibility where necessary.

**Response:** Thank you very much for your suggestion. We have checked every figure and reduced the line thickness of the profiles to make it better visibility.

4) Regarding labels and legends:
- the axes tick labels are sometimes wrong (see e.g. horizontal axes in Figure S19)
- please double check units (see e.g. Fig. 6, where O4 dSCDs are given in molec cm-2)
- Assure readability of legends
- Avoid long legend labels (an extreme case is Figure 16, as discussed in the specific comments)

**Response:** Thank you very much for your suggestions. We have checked every figure and made the corrections.

5) Where possible, consider applying the same horizontal axes limits in different (sub-)plots.
**Response:** Thank you very much for your suggestion. Please note that if the same horizontal axes limits in different subplots were applied, then the results for the low values cannot clearly be recognized by the reader. Thus we did not use the same limits.

2. Some of the wordings in the discussions are not clear to me or at least hard to follow and particularly the final conclusions should be more quantitative. The corresponding paragraphs are listed in the specific comments. A general issue is that authors seem to use the term "systematic deviations" sometimes for the relative differences (hence, considering the sign of the deviations) and sometimes for the general magnitude (independent of the sign) of the systematic deviations. This should somehow be clarified, probably by consistently using the expressions "differences" and "deviation magnitudes" for the first and the second case, respectively.
**Response:** Thank you very much for your suggestion. We used the term "systematic deviations" to describe the general magnitude (independent of the sign) of the deviations, including the relative and absolute differences. In order to make it more clear, we have revised the manuscript according to your suggestions. We now use "differences" and "deviation magnitudes" for the first and the second case, respectively, and revised the manuscript accordingly.
We also followed the suggestion that the final conclusions should be more quantitative, see our reply to the specific comments.

3. The authors state, that their findings "explain part of the deviations between the AOD retrieved from MAX-DOAS and sun photometers in previous studies" (P2L5 but also P24L12). First, I do not agree with the word "explain" here, since the presented results simply show the same behavior as observed in former publications. In fact, an actual "explanation" for these systematics has already been proposed by Irie (2008), Frieß (2016), and Bösch (2018): they proposed that biases introduced by the a priori assumptions are responsible for the deviations. Second, Tirpitz et al. (2021) have shown, that, in the case of OEM algorithms, these biases can be accounted for by applying a "partial AOT correction": by taking AVK smoothing effects into account, they estimate the fraction of the AOD that MAX-DOAS inversions are actually able to perceive, and by applying corresponding correction factors, the AOD underestimation observed for MAX-DOAS inversions can largely be removed. Ideally, a "partial AOT correction" should also be performed for the AODs in the presented manuscript (as it is expected to remove large parts of the discrepancies). If the authors think this is out of the scope of their study, they should at least mention the above publications and corresponding explanations/correction approaches.
**Response:** Many thanks for these suggestions! We now state that our results confirm the results of previous studies (e.g. from Irie et al, 2006, Frieß et al., 2016, Bösch et al., 2018) that part of the deviations between the AOD retrieved from MAX-DOAS and sun photometers can be explained by the biases introduced by the a priori assumptions. We introduced a new sub-section (3.3.) which describes the main findings of our study and relates them to previous studies.
**Changes in manuscript:**
**3.3 Comparison with the earlier studies**
In this section we discuss the most important findings of our investigations and compare them to the

[revised manuscript text omitted]

4. P5L8-11: Hard to understand. Maybe simplify the sentence at this point: "We compare the aerosol and trace gas profile retrieval results from two MAX-DOAS inversion algorithms (PriAM and MAPA, for details, see below) for different aerosol and trace gas scenarios."

**Response:** Thank you for your advice. The text was changed to make it more clear.

**Changes in manuscript:** Here, we compare the aerosol and trace gas profiles retrieved from MAX-DOAS by two inversion algorithms (PriAM and MAPA, for details see below) with the input values (used as input for the DSCD simulations) for different aerosol scenarios. We also investigate the effects of the aerosol extinction and optical properties, including single-scattering albedo (SSA) and the asymmetry parameter (AP), on the aerosols profiles retrieved by PriAM in the UV and Vis.

5. P5L11-12: remove the sentence "For trace gas retrievals…" and improve the explanation in Section 2.1. instead.

**Response:** Thank you for your advice. It was removed in the P5L11-12 and moved to the section 2.1 P6L12-15.

6. Section 2.1: refer to Figure 1 more often in the text to help the reader understand the strategy, e.g.:
P6L1: "A set of atmospheric scenarios (orange box on the very left), ..."
Explain the two strategies here already (please do this carefully also by referring to the Boxes S1 and S2 in Fig.1), such that the reader understands the entire figure before moving on.

**Response:** Thank you for your suggestions.

**Changes in manuscript:** P6: A set of atmospheric scenarios (orange box on the left side), including variations of the viewing geometries, single-scattering albedos, and asymmetry parameters, was used to simulate the SCDs of traces gases and $O_4$, which will be described in detail in Section 2.2. The first step was to quantitatively evaluate the effect of different aerosol loads on the aerosol inversion (The upper part of the Fig.1). For that purpose the simulated $O_4$ DSCDs were used as input for the aerosol profile retrievals. The retrieved and input aerosol profiles were then compared in order to characterize the effect of the aerosol properties (in particular the AODs) on the retrieved aerosol profiles. The second step was to quantitatively evaluate the effect of different aerosol loads on the trace gas inversion (the bottom half of the Fig.1). For the trace gas retrievals, we apply 2 retrieval strategies where either the retrieved (S1, red box in the lower half of Fig.1) or the input (S2, red box in the lower half of Fig.1) aerosol profile is used.

7. P6L5: "SCDs" are not introduced and this is should also not be necessary. Change "SCDs" to "DSCDs" here and for all following occurrences of "SCDs". Further, pay attention to consistency: either "dSCD" or "DSCD" (e.g. P6L19

**Response:** Thank you for your reminding. We checked the full text and changed SCDs to DSCD.

**Changes in manuscript:**

P6: For that purpose the simulated $O_4$ DSCDs were used as input for the aerosol profile retrievals.

P7. The differences of the simulated $O_4$ DSCDs by both models are discussed in section 3.1.2.

P11: The final profiles are weighted averages of the best matching profiles for the given trace gas DSCDs.

P15: Because the aerosol properties used in the MAPA LUT (SSA = 0.95 and AP = 0.68) are different from those used for the simulations of the $O_4$ DSCDs by SCIATRAN (SSA = 0.90 and AP = 0.72), two sets of $O_4$ DSCDs for SSA and AP (SSA = 0.90 or 0.95 and AP = 0.72 or 0.68) were simulated by MCARTIM.

P15: Using McArtim for the calculation of synthetic DSCDs, i.e. consistent RTM in forward model and inversion, results in much better agreement, in particular for low AOD.

P19: Part of the systematic underestimation of the MAPA AODs for exponential profiles is probably caused by the differences of the RTM (SCIATRAN v2.2) and settings (SSA=0.9, AP=0.72) used for the simulation of the input $O_4$ DSCDs and for the MAPA algorithm (MCARTIM, SSA=0.95, AP=0.68), see **Fig. S11**.

8. P6L14: "assumed input profiles" for consistency.

**Response:** Thank you for this suggestion.

**Changes in manuscript:** P6. Before the effects of different aerosol loads on the retrieval of aerosol and trace gas profiles were analyzed, some basic parameters were prescribed for simulating the $O_4$ and trace gas SCDs for the 'assumed input profiles' in the RTM.

9. P7L11: "Section 3.1.2"

**Response:** Thank you for your suggestion. It was changed.

**Changes in manuscript:** P7. The differences between O$_4$ DSCDs simulated by SCIATRAN and MCARTIM are further investigated in Section 3.1.2.

10. P7L16: "wavelengths of 360 nm"

**Response:** Thank you for your suggestion. It was changed.

**Changes in manuscript:** As standard settings we chose wavelengths of 360 nm and 477 nm, elevation angles of 1°, 2°, 3°, 4°, 5°, 6°, 8°, 15°, 30°, and 90° (the same as the settings in the CINDI 2 campaign, Kreher et al., 2020).

11. P7L18: The described profile shapes were generally used for different investigations, not only as a priori, right? Maybe keep this more general and remove "as a-priori".

**Response:** Thank you for your suggestion. You are right. The described profile shapes were generally used for different investigations. We changed the text as suggested.

**Changes in manuscript:** P8

In the real atmosphere, a large variability of aerosol and trace gas profiles exists. However, we had to limit our profile shapes to typical profile shapes, which occur in the atmosphere. In this study, three different profile shapes were used, which are Exponential, Boltzmann, and Gaussian profile shapes:

a) Exponential profiles: such profiles are typical if the emissions mainly occur at the surface. During transport to higher layers the concentration systematically decreases with altitude. The scale height depends on the atmospheric lifetime and the vertical transport time. The description for Exponential functions of altitude z as follows:

Exponential: $f_E(z) = A_E(h_E) \times \exp(\frac{-z}{h_E})$ with scale height $h_E$,

b) Boltzmann profiles: Such profiles represent situations, for which a layer is quickly mixed (compared to the lifetime of the species), and there is a barrier for further upwards transport above that layer. Such situations typically occur for well mixed boundary layers. The description for Boltzmann functions of altitude z as follows:

Boltzmann: $f_B(z) = \dfrac{A_B(h_B)}{1 + \exp(\frac{-(z - h_B)}{0.3})}$ with effective profile height $h_B$.

c) Gaussian profiles: in our study these profiles describe elevated layers. Such profiles represent situations with long range transport of pollutants, which typically occurs above the boundary layer. Elevated profiles might also occur for aerosols and trace gases which are secondary formed, while air is transported upwards. The description for Gaussian functions of altitude z as follows:

Gaussian: $f_G(z) = A_G(h_G, \sigma) \times \exp(\frac{-(z - h_G)^2}{2\sigma^2})$     with peak height $h_G$, and the full width at

half maximum (FWHM) $\sigma$.

12. P8L11: "a priori" instead of "a-priori". Check further occurrences throughout the manuscript.

**Response:** Thank you for your suggestion. We checked the full text and changed "a-priori" to "a priori".

**Changes in manuscript:**

P9:     For RTM calculations, vertical profiles of the aerosol extinction $\varepsilon$ and $NO_2$ concentration c are generated by multiplying f with the respective *a priori* column:

**Table 1** lists the parameters used for RTM, including solar/viewing geometry, a priori AOD/VCD, and parameters for the different profile shapes.

13. P8L16: "were PriAM and MAPA, as listed …"

**Response:** Thank you for your suggestion. It was changed.

**Changes in manuscript:** P9. The retrieval algorithms used in the comparison were PriAM and MAPA, as listed in **Table 2**.

14. P10L17: AODs = 3 are included or? So it should be AODs less-equal 3 instead of less than 3.

**Response:** Thank you for your suggestion. AODs =3 are included. We changed the text as suggested.

**Changes in manuscript:** P11. It is worth noting that the maximum AOD in MAPA is 3, since higher AODs were not included in the RTM look-up table; therefore, only aerosol scenarios with AOD $\leqslant$ 3 were included in this study for MAPA.

15. P11L1: Sentence seems messed up: multiple references to Table 1 and a bracket out of place. Please correct/rephrase.

**Response:** Thank you for your suggestion. We corrected the sentence.

**Changes in manuscript:** P12. In order to limit the number of investigated profiles, first a sensitivity study with PriAM was carried for the selected profile shapes in Table 1 (these best represent the variety of realistic profile shapes).

16. P11L3-8: I do not understand the criteria on which the authors took the decisions here. This needs to be explained in more detail. Currently, I do not see much value in the presented side investigation. Therefore, alternatively, the corresponding investigation might be removed completely, and instead only the four profiles of relevance for the rest of the study might be introduced in the text and in Table 1. This might also avoid some confusion.

**Response:** Thank you for your suggestions. We think this investigation should be retained, although similar conclusions for the same profile shape were obtained. But the results were slightly different for the same profile in different heights. For example, when the scale heights of the exponential profile are low, the retrieved profiles are close to the input profiles. But for high scale heights the retrieved scale heights underestimated the true high scale heights. So we chose two exponential profiles. This information was added to the text.

**Changes in manuscript:** P12: For the exponential profiles, two height parameters were chosen, because for both height parameters systematically different results were obtained: when the scale heights of the exponential profiles are low, the retrieved profiles are close to the input profiles. But for high scale height, the retrieval underestimates the scale height of the exponential profiles.

17. P11L19: "In this Section the effect…"

**Response:** Thank you for your suggestion. We made the correction.

**Changes in manuscript:** P12. In this Section the effect of different AOD on the retrieval of aerosol profiles are presented for a scenario with SZA = 60°, RAA = 120°, SSA = 0.9, and AP = 0.72. Note that similar results were found for different scenarios for both PriAM and MAPA.

18. P13L8-11: I cannot follow. On the one hand, the relative deviations (I guess it should be the "relative deviation magnitude") increases with AOD but then it does not? Please clarify.

**Response:** Thank you for your remark. The sentence "But the relative deviation magnitude does not increase with the increase in AOD." was removed.

19. P14L17: Also add the scale height. It might be useful to add a "default a priori profile"-row to Table 1 with the relevant properties.

**Response:** Thank you for your suggestion. We added the new Table 4, with information about the *a priori* profile and the *a priori* covariance.

**Changes in manuscript:** P16. Here it should be noted that an exponential shape with an AOD of 0.2 and a scale height of 1.0km was used as universal *a priori* profile in this study.

**Table 4. Parameter settings used in the general PriAM retrieval for aerosol and NO₂ profiles.**

| Parameters | | |
|---|---|---|
| a priori profile | Aerosol : | exponential shape with an AOD of 0.2 and the scale height of 1.0km |
| | NO$_2$: | exponential shape with the VCD of $1.0 \times 10^{15}$ molecules cm$^{-2}$ and the scale height of 1.0km |
| Sa_ratio | Aerosol : | 0.1 |
| | NO$_2$: | 0.5 |

20. P15L1-3: It might be interesting to briefly discuss the motivation for this approach and particularly its relevance for real measurements, where there is basically no information on the vertical distribution or the AOD prior to the inversion. Do have any strategy in mind on how real retrievals might be improved e.g. by iteratively adapting the a priori profile (which is btw. quiet arguable as it violates the OEM principle)? It might be worth discussing this at least at some point (Conclusions?)

**Response:** Thank you for your suggestion. A discussion of the implications of this study for real observations is added at the end of the paragraph.

**Changes in manuscript:** P16. In order to investigate the importance of the *a priori* profile for the aerosol profile retrieval, the influence of the *a priori* profile was analyzed by changing the *a priori* profile to different aerosol profile shapes.

P17. This provides a possibility for real measurements to obtain more accurate aerosol profiles if independent information on the *a priori* profiles is available, e.g. from Lidar observations and sun photometers.

21. P15L8-9: "no effect" doesn't seem right here. According to Figure 4, for the Boltzmann and the Gaussian input profiles it has "little effect", while for the exponential input profiles there are significant differences right? Please clarify.

**Response:** Thank you for your suggestion. The text was corrected.

**Changes in manuscript:** P16. However, increasing the AOD of the universal (exponential) *a priori*

profile exhibited only little effect on the inversion results of the Boltzmann and Gaussian shapes.

22. Section 3.1.3: What are the values of the off-diagonal elements of Sa? Btw. the default a priori covariance information also be included in Table 1.

**Response:** Thank you for your suggestion. The values of the off-diagonal elements of Sa are the square of the a priori profile. In order to better explain Sa, a new symbol (Sa_ratio) is introduced. We added the new Table 4 (see above), which contains the description of the *a priori* profile and the *a priori* covariance.

**Changes in manuscript:** P17

The Sa is the covariance matrix of the *a priori* profile (N×N), and its diagonal elements are the square of the *a priori* state uncertainties with the off-diagonal elements calculated from the Gaussian function with the correlation length of 0.5 km (Frieß et al., 2006).

The diagonal elements of Sa for the aerosol profile were set as the square of the *a priori* profile uncertainty. The standard settings for the *a priori* profile uncertainty were 10% of the *a priori* profile. To describe this ratio, a new symbol (Sa_ratio) is introduced (see Table 4).

**Table 4. Parameter settings used in general PriAM retrieval for Aerosol and NO₂ profiles.**

| Parameters | | |
|---|---|---|
| a priori profile | Aerosol : | exponential shape with an AOD of 0.2 and the scale height of 1.0km |
| | NO$_2$: | exponential shape with the VCD of $1.0 \times 10^{15}$ molecules cm$^{-2}$ and the scale height of 1.0km |
| Sa_ratio | Aerosol : | 0.1 |
| | NO$_2$: | 0.5 |

23. P16L1: what does "correlation" mean here? Correlation coefficient?

**Response:** Thank you for your remark. It was the correlation coefficient.

**Changes in manuscript:** P17. For the exponential profiles with a scale height of 0.5 km, the correlation coefficient between the retrieved and input aerosol profiles decreased with increasing Sa.

24. P16L5-6: Shouldn't this be the other way round? Also, it is not ideal to talk of "limits" in the context of OEM approaches. I propose: "This is due to the fact that biases towards the a priori profiles are reduced with increasing Sa values."

**Response:** Thank you for your suggestion. We changed the text accordingly.

**Changes in manuscript:** P18. In particular, the retrieved surface extinctions and scale heights could be improved by increasing the Sa. This is due to the fact that biases towards the *a priori* profiles are reduced with increasing Sa values.

25. P16L7: Give an approximate altitude for "upper layers".

**Response:** Thank you for your suggestion. The approximate altitude for "upper layers" was above 2.0 km.

**Changes in manuscript:** P17. When the Sa values were too large, however, the retrieved aerosol profiles in the upper layer (approximately above 2.0 km) were more unstable.

26. P16L8-11: In my opinion, this is the major finding of the section. However, it only considers the correlation coefficient. What about the actual agreement (e.g. in terms of RMSD). This is probably

the most important quantity to minimise.

**Response:** Thank you for your advices. We calculated the RMSD according to your suggestions. We found that the RMSD was the smallest when the correlation coefficient was the highest. And we also quantified the RMSD in the manuscript.

**Changes in manuscript:** P18. The highest correlation coefficient was found when the diagonal elements of Sa were set to the square of 20% of the *a priori* profile for the Boltzmann profiles and exponential profiles with a scale height of 1.0 km at AOD of 5.0, with the smallest root-mean-square deviation (RMSD) of 0.54 and 0.50 (averaged of 360nm and 477nm for each shape), respectively. For the Gaussian profile, the correlation coefficient was highest with the diagonal elements of Sa in 50% of the *a priori* profile. The smallest averaged RMSD of 0.55 was also found for this scenario with values of 0.58 at 360nm and 0.52 at 477nm, respectively.

27. P16L17: AOD less-equal than 3 (?)

**Response:** Thank you for your question. The AOD is less-equal than 3. We changed the text accordingly.

**Changes in manuscript:** P18.   Note that only the results for AOD $\leqslant$ 3.0 were derived from MAPA. Also the slopes, intercepts, and correlation coefficients are shown in **Fig. 7**.

28. P17L8: Remove double full stop.

**Response:** Thank you for this hint. It was removed.

29. P17L13: comma after "AOD"

**Response:** Thank you for this hint. The comma was added after AOD.

**Changes in manuscript:** P19. Especially for low AOD, the AODs retrieved by PriAM are closer to the input AODs than those retrieved by MAPA.

30. P17L14-19: I would expect the different ways of how a priori information is incorporated in the two retrievals as a major reason: for PriAM this is of course the a priori profile and the a priori covariance. For MAPA a priori assumptions are incorporated in the form of prescribed profiles described few parameters. This might be added as another potential reason.

**Response:** Thank you for your advice. The information was added.

**Changes in manuscript:** P19. The different incorporated methods for providing the a priori information is also a potential reason for the differences between the two retrieval algorithms. Prescribed a priori profile and the *a priori* covariances are used in PriAM, while a priori assumptions are incorporated in MAPA in the form of prescribed profile shapes by the chosen parameterization.

31. P18L5: Would be helpful to give the applied values for SAA and AP in brackets here again.

**Response:** Thank you for your suggestion. It was added.

**Changes in manuscript:** P20. First, a single aerosol profile was used to simulate the $O_4$ DSCDs for different SSA (0.8, 0.9, 1.0) and AP (0.68, 0.72) values (See **Table 1**).

32. P18L7: "…using the "correct" SSA and AP values (hence, the same values as they were applied in the corresponding O4 DSCD simulations)"

**Response:** Thank you for your suggestion. We changed the text accordingly.

**Changes in manuscript:** P20. Next, the simulated $O_4$ DSCDs were used to retrieve the aerosol extinction profiles by PriAM using the "correct" SSA and AP values (hence, the same values as they were applied in the corresponding $O_4$ DSCD simulations).

33. P19L1: What do the numbers "0.01 to 1.5" represent? I guess these are absolute deviation magnitudes in the extinction coefficient? Please clarify and add units if necessary.

**Response:** Thank you for your remark. The umbers "0.01 to 1.5" represent the absolute deviations of the extinction coefficient. This information was added to the text.

**Changes in manuscript:** P21. The effect of incorrect SSA and AP values on the aerosol profiles retrieved by PriAM increased with increasing AOD with the absolute deviations of the extinction coefficient increasing from 0.01 to 1.5 $km^{-1}$ as the AOD increased from 0.1 to 5.0.

34. P19L5: Add the 5 VCD values in brackets here again.

**Response:** Thank you for your suggestion. It was added.

**Changes in manuscript:** P21. First, the effects of different aerosol extinction profiles on the trace gas profile inversion for 5 $NO_2$ VCDs ($0.1\times10^{16}$, $0.3\times10^{16}$, $1.0\times10^{16}$, $3.0\times10^{16}$, and $10.0\times10^{16}$ molecules $cm^{-2}$) were examined using aerosol profiles with 4 AODs (0.3, 1.0, 3.0, and 5.0) (AOD = 5.0 was not included for MAPA).

35. P19L6: Add one sentence here again on "S1" and "S2" to remind the reader of approximate meaning.

**Response:** Thank you for your suggestion. The information was added.

**Changes in manuscript:** P21. Two strategies (either the retrieved (S1) or the input (S2) aerosol profiles served as input for the retrievals of the $NO_2$ profiles) were employed to retrieve the $NO_2$ profiles (see **Section 2.1**).

36. P19L21: Shouldn't this be "Fig 10" instead of "Fig. S10"?

**Response:** Thank you for this hint. All the figures were changed due to the suggested changes (the retrieved profiles are now shown in the main text and the relative deviations in the supplementary material). Thus also all figure numbers in the manuscript are changed.

37. P19L19: "…, the magnitude of absolute deviations between the retrieved …"

**Response:** Thank you for your suggestion. The text was changed accordingly.

**Changes in manuscript:** P22. For the same aerosol conditions, the magnitude of the absolute deviations between the retrieved $NO_2$ profiles and the input values increase with increasing $NO_2$ VCDs.

38. P19L20: "… with increasing NO2 VCDs. However, the relative deviations…"

**Response:** Thank you for your suggestion. The text was changed accordingly.

**Changes in manuscript:** P22. However, the magnitude of the relative deviations stays constant (**Fig. S20**).

39. P19L19 – P20L1: I would say, the relative deviations generally increase with AOD, don't they?

**Response:** Thank you for this hint. The content of P19L19-P20L1 is to introduce the effect of $NO_2$

VCD on inversion. But the effect of AOD is exactly what you said. The relative deviations magnitude generally increase with AOD for the same $NO_2$ VCD. It was added at the end of P20L1.

**Changes in manuscript:** P22. For the same aerosol conditions, the systematic deviations between the retrieved $NO_2$ profiles and the input values increase with increasing $NO_2$ VCDs, while the magnitude of the relative deviations increases slightly with the increase of AOD for the same $NO_2$ VCD.

40. P20L1: What is meant by "The systematic deviations here"? The largest deviation magnitudes? Absolute or relative? Please clarify.

**Response:** Thank you for your remark. It means the largest deviation magnitudes. We changed the text accordingly.

**Changes in manuscript:** P22. The largest deviation magnitudes between the retrieved $NO_2$ profiles and the input $NO_2$ profile for the exponential $NO_2$ profiles with scale height of 0.5 km were mainly found below 1.0 km. The largest deviation magnitudes between the retrieved $NO_2$ profile and the input $NO_2$ profile appeared below 2.0 km for the other three profile shapes, with the maximum deviation magnitude occurring at 1.0 km and 0.2 km.

41. P20L8-13: Do these findings apply for both algorithms or only for PriAM? Please clarify.

**Response:** Thank you for your suggestion. These findings apply only for PriAM. This was made clear in the text.

**Changes in manuscript:** P22. The smoothing effect of PriAM overestimates the $NO_2$ concentrations around 500 m to compensate for the underestimation of the $NO_2$ concentrations above 1.0 km. In other words, PriAM yields another solution for the ill-conditioned problem in order to achieve convergence between the retrieved and measured SCDs under the control of the *a priori* profile and its covariance.

42. P20L13 Remove either "uncertainty" or "covariance"

**Response:** Thank you for your suggestion. We changed the text accordingly.

**Changes in manuscript:** P22. In other words, the PriAM yields another solution for the ill-conditioned problem in order to achieve convergence between the retrieved and measured SCDs under the control of the *a priori* profile and its covariance.

43. P20L21: Start a new paragraph before "The NO2 profiles…"

**Response:** Thank you for your suggestion. We added a new paragraph.

44. P21L4-6: What is meant by "singular values". Outliers in single layers? I do not see such things in the figures. Please clarify.

**Response:** Thank you for your remark. The "singular values" mean the outliers in some layers, which obviously deviated from the true values. This information was added to the text.

**Changes in manuscript:** P23. An interesting phenomenon was the occurrence of some singular values (outliers which deviate from the true values in some layers) in the upper layers of the retrieved profiles for low $NO_2$ VCDs (mainly for $NO_2$ VCD $< 1 \times 10^{16}$ molecules cm$^{-2}$).

45. P21L17: "…value of the Sa diagonal…"

**Response:** Thank you for this hint. Sa was added.

**Changes in manuscript:** P24. As standard value of the Sa diagonal elements for retrieval of $NO_2$

profiles, we used the square of 50% of the *a priori* profile.

46. P23L5-8: Similarly as for the AOD (see general comments) I would suspect a priori biases as the reason for the systematic deviations. It might be out of the scope of the study, but I encourage the authors to try a corresponding correction: Convolute the input profiles with the retrieval AVKs, recalculate the VCD from the smoothed profile, and compare this "a priori bias-corrected" true VCD to the retrieved VCDs. Do the systematic deviations disappear?

**Response:** Thank you for your suggestion. Concerning your suggestion about convoluting the input profiles with the retrieval AVKs, we did not follow this suggestion.

The main reason is that the spatial resolution of the input profile (<200m) and retrieved AVKs (200m) is different. So it can't convolve directly.

47. P23L20: add VCD range here
**Response:** Thank you for your suggestion. The information was added.
**Changes in manuscript:** P30. In addition, a series of $NO_2$ scenarios was assumed with the same profile shapes and various VCD values (from $0.1 \times 10^{16}$ to $10.0 \times 10^{16}$ molecules cm$^{-2}$).

48. P24L5: Please provide at least some order of magnitude or a range of deviations.
**Response:** Thank you for your suggestion. The information was added.
**Changes in manuscript:** P30. However, for most cases the deviations caused by wrongly assumed AP and SSA were found to be rather small compared to other uncertainties. The maximum relative deviation was generally found around 1.0km with the values of about 25%.

49. P24L12-14: See general comment regarding systematic deviations between MAX-DOAS and sunphotometer observations. Please discuss the actual reasons here and give credits to the corresponding former publications.
**Response:** Thank you for your suggestion. The information was added.
**Changes in manuscript:** P31. Such a systematic underestimation has also been found in several previous studies (eg. Irie et al., 2008, Frieß et al., 2016, Bösch et al. 2018, and Tirpitz et al., 2021). The systematic deviation between MAX-DOAS and sun photometers is partly caused by the missing sensitivity of MAX-DOAS observations for higher altitudes, especially for optimal estimation algorithms.
We also added a new sub-section (3.3), see above.

50. P24L17: "…in the RTM model. It…"
**Response:** Thank you for your hint. We changed the text accordingly.
**Changes in manuscript:** P31. For MAPA, part of the differences between input and retrieved AODs can be explained by the differences in the RTM model.

51. P25L4-6: This is very likely due to the corresponding reduction of a priori biases. Might be added as a potential explanation.
**Response:** Thank you for your suggestion. We changed the text accordingly.
**Changes in manuscript:** P31. The main reason is probably that the corresponding *a priori* bias was reduced.

52. P26L7: What does "single outliers" mean. Single profiles? Single layers? Please explain further.
**Response:** Thank you for your remark. It means single outliers in same layers. We changed the text

accordingly.

**Changes in manuscript:** P33. The increase of the Sa values did not improve the inversion results for high AODs, but instead lead to the occurrence of single outliers in some layers.

53.  Figure1: Change upper green diamond to "Comparison of "input" and retrieved aerosol profiles".

Change lower green diamond to "Comparison of "input" and retrieved trace gas profiles"

**Response:** Thank you for your suggestion. We changed the figure accordingly.

**Changes in manuscript:**

[Figure]

54.  Figure 2 caption: "where the retrieved AOD exceeds 2". Shouldn't this be "3"?

**Response:** Thank you for this hint. We changed the text accordingly

**Changes in manuscript:**

**Figure 2. Comparison of the aerosol profiles retrieved by PriAM and MAPA for 360 nm (first line) and 477 nm (second line) and the corresponding input aerosol profiles for (a) exponential shape with h = 0.5 km, (b) exponential shape with h= 1.0 km, (c) Boltzmann shape, and (d) Gaussian shape.**

**The red and blue curves indicate the results from PriAM and MAPA, respectively. The corresponding relative deviations and absolute deviations are shown in Fig. S8 and Fig. S9, respectively. Note that MAPA by default flags cases where the retrieved AOD exceeds 3, thus the high aerosol scenarios are missing for MAPA.**

55.  Figure 5: Legend: add a square root over "Sa" or a square to the numbers ($0.06^2$, $0.1^2$, …).

**Response:** Thank you for your suggestion. The Sa_ratio is introduced in the question 22. Here we used the Sa_Ratio.

**Changes in manuscript:**

[Figure]

56. Figure 6: Units should be molec$^2$ cm$^{-5}$

**Response:** Thank you for this hint. We changed the text accordingly

**Changes in manuscript:**

[Figure]

57. Figure 7, legend: remove typo "deviatiobs"

**Response:** Thank you for this hint. It was corrected

**Changes in manuscript:**

[Figure]

58. Figure 10, end of caption: "…shown at the top."

**Response:** Thank you for this suggestion. We changed the text accordingly.

**Changes in manuscript: (now Figure 11)**

**Figure 11. Retrieved NO₂ profiles by PriAM and MAPA for scenario S1 (see text) for aerosol profiles with 3 selected AODs (0.3, 1.0, and 3.0) and 5 NO₂ VCDs for of (a) exponential shape with**

**The solid and dotted colored lines refer to the AODs and algorithms shown at the top**

59. Figure 13: Might be useful to add the default a priori uncertainty as error bars or shaded area.
**Response:** Thank you for your suggestion. The a priori uncertainty was added.
**Changes in manuscript:**

[Figure]

60. Figure 16: Legend is extremely bulky making it harder for the reader to understand the figure. One column might be enough since absolute and relative deviations are shown in separate subplots. Furthermore, the legend might be equipped with a title like "Deviations between retrieved and input NO2 VCDs". Labels can then be reduced to something like "Priam_S1, AOD 0.3".
**Response:** Thank you for your suggestion. The figure was revised accordingly.
**Changes in manuscript:**

[Figure]

Thank you for taking care of our manuscript.

Kind regards,
Xin Tian
E-mail: xtian@ahu.edu.cn

Corresponding author: Yang Wang, Pinhua Xie,
E-mail address: y.wang@mpic.de; phxie@aiofm.ac.cn;

---

## Author Comment (AC2)

Dear editors and reviewers,

Thank you very much for your constructive comments and advices on our manuscript. Your positive evaluation and comments encourage us and are a great help for us. We have carefully considered every comment, and made the corresponding revisions in the revised manuscript (indicated by the 'tracked changes').

Point to point response is following:

**Major comments**

1) My main criticism of this study is that it shows literally hundreds of figures without coming to clear conclusions. This is not the first study of its kind, so the main question is: What are new and interesting results of this study not yet published elsewhere, and how can these new results be understood?

From what I understood, the main results are:

- Aerosol parameters SSA and asymmetry factor are not as critical as one may have thought for the aerosol retrieval
- Changing the covariance matrix changes the results of the OE retrieval as it results in different weighting of a priori and measurements in the inversion
- NO2 profiles are not very sensitive to the aerosol profiles used
- AOD is systematically underestimated by MAX-DOAS retrievals
- Low NO2 columns are overestimated, high NO2 columns are underestimated

The first four points have already been discussed in the literature before but maybe not with this level of detail. The last one is new to me and would deserve more discussion as it is unexpected and surprising. What could be the reason for such a behaviour?

**Response:** Thank you for your suggestion. We reorganized the abstract and conclusion to make the results clearer.

**Changes in manuscript:**

[revised manuscript text omitted]

2) As this study is on synthetic data which are necessarily idealized in many ways, the question is: Which of these results are of relevance for real MAX-DOAS measurements? Are there any take-home messages for people working on MAX-DOAS profiles? What is specific to the two inversion codes used, what is fundamental to MAX-DOAS retrievals?

**Response:** Thank you for this comment.

In our opinion, all findings of our study are relevant for real measurements. We summarized them now in a clearer way in the conclusions of the updated manuscript. We also added a new section (3.3) with the comparison of our results to previous studies (see next point).

3) In general, I think a section on comparison of the results found here with what was reported in earlier studies should be added.

**Response:** Thank you for your suggestion. The comparison to the findings of previous studies was added.

**Changes in manuscript:**

3.3 **Comparison with the earlier studies**

[revised manuscript text omitted]

4) Something I could not find in this manuscript is information on the uncertainties assumed for the slant columns. I assume that no noise was added to the results from the RTM but still the retrievals must have made an assumption on the uncertainties. This is an important point which needs to be added to the manuscript as it can have a large impact on the results.

**Response:** Thank you very much for your remark. It clarified our assumptions in P11L22-P12L1 to make it clear.

**Changes in manuscript:** The fitting error for all $O_4$ DSCDs is set as $0.03\times10^{43}$ molecules$^2$ $cm^{-5}$, and that for $NO_2$ DSCDs to 1% of the $NO_2$ DSCDs in the PriAM and MAPA retrievals.

5) Another information I'm missing is what the atmosphere in the forward simulations looked like above 4 km. Was there any NO2 or aerosol present at higher altitudes as well?

**Response:** Thank you very much for your questions. The value above 4.0 km is set to 0. And it is added in the P9L4 to make it clear. In the real atmosphere, aerosols and gases are typically concentrated below 3 kilometers (or even lower).

**Changes in manuscript:** P9L4. The value above 4 km altitude is set to 0.

6) Throughout the manuscript, results are shown for two wavelengths, but there is no discussion whatsoever of similarities and differences between these results. If there is no discussion then I do not see the reason for adding all these figures.

**Response:** Thank you very much for your suggestions. We have added a discussion of the similarities and differences between the two wavelengths to the abstract and conclusions.

**Changes in manuscript:**

**Abstract**:

Interestingly, the results for both investigated wavelengths (360 nm and 477 nm) were found to be rather similar indicating that the differences in the radiative transfer between both wavelengths have no strong effect.

P18. The highest correlation coefficient was found when the diagonal elements of Sa were set to the square of 20% of the *a priori* profile for the Boltzmann profiles and exponential profiles with a scale height of 1.0 km at AOD of 5.0, with the smallest root-mean-square deviation (RMSD) of 0.54 and 0.50 (averaged of 360nm and 477nm for each shape), respectively. For the Gaussian profile, the correlation coefficient was highest with the diagonal elements of Sa in 50% of the *a priori* profile. The smallest averaged RMSD of 0.55 was also found for this scenario with values of 0.58 at 360nm and 0.52 at 477nm, respectively.

P33 Finally it should be mentioned that the results of this study are very similar for both selected wavelengths (360 and 477 nm) indicating that the differences in the radiative transfer between both wavelengths have no strong effect on the MAX-DOAS profile retrievals.

7) The authors decided to put the figures showing relative differences in the manuscript and the other figures in the supplement. I'd suggest to do the opposite and to show the retrieved profiles in the main text, adding the true and the a priori profiles. In my opinion, these figures give a more rapid access to the performance of the retrievals while the relative differences are additional information, which can be moved to the supplement.

**Response:** Thank you very much for your suggestion. We have considered your advice to change the figures in the main text. The actually retrieved profiles were moved to the main text, and the relative deviation was moved to the supplementary material. And we also changed the corresponding content in the article.

8) I found it a bit unfortunate that the authors decided not to include a perfect scenario, where the profile shape and AOD of the a priori agree with the true profile. It would be very interesting to see, if in this case PRIAM also underestimates the AOD / NO2.

**Response:** Thank you very much for your suggestions. We have considered your advice and added the correlative sensitivity analysis. The corresponding result for aerosol and gas profile were added in Sec.3.1.3 P16 and Sec. 3.2.1 P25, respectively.

**Changes in manuscript:**

**3.1.3**

P16 We also investigated the retrieval results if exactly the a priori profiles were used as input. The results are presented in **Fig. 5**. The results show that the retrieved aerosol profiles are basically the same as the input profiles, and the relative deviation is less than 0.05% (**Fig. S14** of the supplement). This

sensitivity study shows that a) PriAM is implemented in a proper way and b) improved retrieval results can be obtained with improved *a priori* profiles. This provides a possibility for real measurements to obtain more accurate aerosol profiles if independent information on the *a priori* profiles is available, e.g. from Lidar observations and sun photometers.

**3.2.1**

P25 We also investigated the retrieval results if exactly the a priori profiles were used as input profiles. The results are presented in **Fig. 16**. In contrast to the aerosol inversion, here for some scenarios substantial differences are found, which in general increase with increasing $NO_2$ VCD and AOD. The smallest deviations are found for exponential and Boltzmann profiles, whereas for Gaussian profiles larger differences are found. The magnitude of the relative deviation increases from 20% to 50% with the $NO_2$ VCD increasing from $1\times10^{14}$ to $10\times10^{16}$ molecules cm$^{-2}$ (**Fig. S28**). It is important to note that the relative deviations for the retrieved $NO_2$ profile by using both the aerosol and $NO_2$ *a priori* profiles as input profiles are less than those if only the aerosol a priori profile is used as input profile (PriAM by S2). This finding also provides guidance for gas inversions in the real atmosphere, if the aerosol and gas profiles can be provided as the *a priori* profile by other monitoring techniques, the inversion results of MAX-DOAS will be more accurate.

Detailed comments

1. Abstract: It is claimed that the finding of the AOD underestimation in the sensitivity study explains the underestimation seen in real data. I think this is neither new, nor an explanation – the explanation as far as I see it is the insensitivity to the upper part of the extinction profile in combination with the forcing of the profile shape from a priori or parametrisation.

**Response:** Thank you very much for your suggestions. We reorganized the abstract, see above.

2. Page 11: The selection of profiles to be used later appears completely random – at least from the text, it is not clear how the "representative" profiles have been selected.

**Response:** Thank you for your remark. On page 12, we introduced the reasons for choosing these "representative" profiles in detail.

**Changes in manuscript:**

In order to limit the number of investigated profiles, first a sensitivity study with PriAM was carried for the selected profile shapes in Table 1 (these best represent the variety of realistic profile shapes). Based on the result shown in **Figs. S2 to S4** it turned out that one height parameter is mostly representative for the parameterization with Gaussian and Boltzmann profiles. For the exponential profiles, two height parameters were chosen, because for both height parameters systematically different results were obtained: when the scale heights of the exponential profiles are low, the retrieved profiles are close to the input profiles. But for high scale height, the retrieval underestimates the scale heights of the exponential profiles.

3. Page 11: The selection of the scenario used for evaluation of the sensitivity to aerosol parameters

could be critical. Have other relative azimuth angles be evaluated as well? I would have expected the effect of the asymmetry factor to be different for different scattering and relative azimuth angles.

**Response:** Thank you for your questions. In this study, we studied the effects of the relative azimuth and solar zenith angles. We found that the results for different SZA (20°, 40°, 60°, 80°) and RAA (30°, 60°, 120°, 180°) are basically the same. But here it is important to note that in the real atmosphere, very different phase functions might occur, and especially for small RAA stronger systematic deviations might occur.

**Changes in manuscript:** P13.

The effects of the different SZA (20°, 40°, 60°, 80°) and RAA (30°, 60°, 120°, 180°) are basically the same. But here it is important to note that in the real atmosphere, very different phase functions might occur, and especially for small RAA stronger systematic deviations might occur. Here only the result for SZA = 60° and RAA = 120° was shown.

4. Page 11: Which aerosol model has been used?

**Response:** We are not sure if we correctly understand this question. Probably you refer to the aerosol phase function. Here we used a HG parameterization. This information was made more clear in the manuscript.

5. Page 15: Which 4 diagonal elements of Sa are you talking about? I assume there are 20 or 21 diagonal elements in Sa? Do the relative values of the diagonal elements in Sa not depend on altitude?

**Response:** Thank you for your remark. The values of the diagonal elements in Sa depend on the *a priori* profile. In other words, they depend on altitude. The description in the article was probably a little unclear. In order to make it more clear, a new symbol (Sa_ratio) is introduced.

**Changes in manuscript:**

P17.    The Sa is the covariance matrix of the *a priori* profile (N×N), and its diagonal elements are the square of the *a priori* state uncertainties with the off-diagonal elements calculated from the Gaussian function with the correlation length of 0.5 km (Frieß et al., 2006).

The diagonal elements of Sa for the aerosol profile were set as the square of the *a priori* profile uncertainty. The standard settings for the *a priori* profile uncertainty were 10% of the *a priori* profile. To describe this ratio, a new symbol (Sa_ratio) is introduced (see Table 4). The 4 Sa_ratio were set to 6%, 10%, 20%, and 50%.

6. Page 16, Line 6: "the higher the Sa values, the lower the upper limits are for the inversion" – this is not clear to me.

**Response:** Thank you for your remark. We added the missing information.

**Changes in manuscript:** P17. This is due to the fact that the biases towards the *a priori* profiles are reduced with increasing Sa values.

7. Page 16, line 22: must be related to systematic performances … or RTM differences

**Response:** Thank you for your suggestion. We changed the text accordingly.

**Changes in manuscript:** P18.    Therefore, it can be concluded that the discrepancies of the retrieved aerosol profiles from the input profiles were not caused by failed convergences of the retrievals but must

be related to systematic performances of the inversion algorithms in solving the ill-conditioned problem or RTM differences.

8.  Page 20, Line 8: "The artificial smoothing effect of the profile inversion algorithm mistakenly overestimates" =>    "The smoothing effect of the profile inversion algorithm overestimates"

**Response:** Thank you for your suggestion. We changed the text accordingly.

**Changes in manuscript:** P22. The smoothing effect of PriAM overestimates the $NO_2$ concentrations around 500 m to compensate for the underestimation of the $NO_2$ concentrations above 1.0 km.

9.  Summary: "We found that both algorithms can reasonably retrieve the 4 aerosol profile shapes" – I'm not sure that readers will agree to this point after having studied the figures with the results. It is clear that the retrievals cannot retrieve the extinction profiles above 1.5 km, and at low and high AOD, they also fail in the lower altitudes for many scenarios.

**Response:** Thank you for your suggestion. We changed the text accordingly.

**Changes in manuscript:** P30. We found that both algorithms have systematic deficiencies in retrieving the 4 profile shapes. Especially at low (above 0.2 km) and high (above 1.5 km) altitudes, often deviations from the true values are found, while for altitudes in between best agreement is found. The algorithms can reasonably retrieve the 4 aerosol profile shapes of AODs < 1.0 for two wavelengths, but for AODs > 1.0 the retrieved values systematically underestimate the true AODs.

10.  Table 1: Why are there stars for both 0.5 and 1.5 km exponentials?

**Response:** Thank you for your remark. For the exponential profile inversions, two exponential profiles are used by default with scale heights of 0.5km and 10km, respectively. So both 0.5 and 1.0 km exponential profiles were marked with stars.

11.  Table 2: not needed

**Response:** Thank you for your suggestion. We have considered your suggestion, but we think that the Table 2 should be retained. It allows the reader to quickly see the differences between the two algorithms.

12.  Figure 2: There is confusion about MAPA excluding scenarios with AOD 2 – please check

**Response:** Thank you for your remark. The AOD is 3. The text was corrected accordingly

**Changes in manuscript: Note that MAPA by default flags cases where the retrieved AOD exceeds 3, thus the high aerosol scenarios are missing for MAPA.**

13.  Figure 7: Typo "deviatiobs"

**Response:** Thank you for your hint. It was corrected

**Changes in manuscript:**

[Figure]

14. Figure 16: It looks as if the bars of the lower 2 lines are partially clipped – please check and change scale if needed

**Response:** Thank you for your suggestion. The changed the scale as suggested.

**Changes in manuscript:**

[Figure]

Thank you for taking care of our manuscript.

Kind regards,

Xin Tian

E-mail: xtian@ahu.edu.cn

Corresponding author: Yang Wang, Pinhua Xie,
E-mail address: y.wang@mpic.de; phxie@aiofm.ac.cn;

---

## Author Response (AR2)

Dear editors,

Thank you very much for your advices. We have carefully revised manuscript and marked every change in red.

The changing in the manuscript as follows:

1. New section 3.3: you should call it "Discussion"
**Response:** Thank you very much for your suggestion. It was changed in the manuscript.
**Changes in manuscript:** P26L19    **3.3 Discussion**

2. P27L1: "They used less profile shapes". I don't understand. Please rephrase
**Response:** Thank you very much for your advice. It was changed to make it clear.
**Changes in manuscript:** P27L1
But compared to this study, they used less scenario profile shapes (Bösch et al. 2018) or they restricted their investigations to a set of profiles with fixed combinations of shapes and vertically integrated quantities (VCDs and AOD).

3. P27L10: After "the most important findings are:", you should itemize your results.
**Response:** Thank you very much for your advice. It was changed to make it clear.
**Changes in manuscript:** P27, P28, and P29

The most important findings are:

[revised manuscript text omitted]

4. P16 and P15: "... the retrieval results if exactly the a priori profiles ...". You should say instead "... the retrieval results in a perfect scenario in which the a priori profile agrees with the true profile"

**Response:** Thank you very much for your suggestion. It was changed in the manuscript.

**Changes in manuscript:** P16L21-22

We also investigated the retrieval results in a perfect scenario in which the *a priori* profile agrees with the input profile.

Thank you for taking care of our manuscript.

Kind regards,
Xin Tian
E-mail: xtian@ahu.edu.cn

Corresponding author: Yang Wang, Pinhua Xie,
E-mail address: y.wang@mpic.de; phxie@aiofm.ac.cn;